# DeepKymoTracker: A tool for accurate construction of cell lineage trees for highly motile cells

Khelina Fedorchuk[1]*, Sarah M. Russell[1,2,3], Kajal Zibaei[1], Mohammed Yassin[1], Damien G. Hicks[1]*

1 Optical Sciences Centre, Swinburne University of Technology, Hawthorn, Victoria, Australia, 2 Immune Signalling Laboratory, Peter MacCallum Cancer Centre, Melbourne, Victoria, Australia, 3 Sir Peter MacCallum Department of Oncology, The University of Melbourne, Melbourne, Victoria, Australia

* kfedorchuk@swin.edu.au (KF); dghicks@swin.edu.au (DGH)

## Abstract

Time-lapse microscopy has long been used to record cell lineage trees. Successful construction of a lineage tree requires tracking and preserving the identity of multiple cells across many images. If a single cell is misidentified the identity of all its progeny will be corrupted and inferences about heritability may be incorrect. Successfully avoiding such identity errors is challenging, however, when studying highly-motile cells such as T lymphocytes which readily change shape from one image to the next. To address this problem, we developed DeepKymoTracker, a pipeline for combined tracking and segmentation. Central to DeepKymoTracker is the use of a seed, a marker for each cell which transmits information about cell position and identity between sets of images during tracking, as well as between tracking and segmentation steps. The seed allows a 3D convolutional neural network (CNN) to detect and associate cells across several consecutive images in an integrated way, reducing the risk of a single poor image corrupting cell identity. DeepKymoTracker was trained extensively on synthetic and experimental T lymphocyte images. It was benchmarked against five publicly available, automatic analysis tools and outperformed them in almost all respects. The software is written in pure Python and is freely available. We suggest this tool is particularly suited to the tracking of cells in suspension, whose fast motion makes lineage assembly particularly difficult.

## Introduction

Time-lapse microscopy measurements have long been used to record the properties, movement and interactions of single cells [1,2]. A variety of new imaging and analysis techniques have been developed to improve the accuracy of tracking and segmentation [2,3]. A particularly demanding application of time-lapse microscopy is to construct cell lineage trees [4–9]. This involves tracking and characterizing a population of dividing cells descended from a common ancestor and constructing the tree of family relationships.

**Data Availability Statement:** The code for tracking cells and for training neural networks, the demo data, the pre-trained weights of the neural networks, and instructions for installation and use

of the code are available at https://github.com/khelina/T-cell-lineages-tracking.

**Funding:** This work was supported in part by National Health and Medical Research Council (NHMRC) grant 2013058. KF and KZ were supported in part by Swinburne University Postgraduate Research Awards (SUPRA), SMR was supported in part by NHMRC grants 620500 and APP1099140 and ARC grant FT0990405, and DGH was supported in part by Australian Research Council (ARC) grant FT140101104. The sponsors or funders had no role in the study design, data collection and analysis, decision to publish, or preparation of the manuscript. ARC - https://www.arc.gov.au/ NHMRC - https://www.nhmrc.gov.au/.

**Competing interests:** The authors have declared that no competing interests exist.

The cell lineage tree is a valuable tool for mapping out the processes of fate determination in a dividing cell system [10,11]. Using this tree, together with information from the segmented cells such as fluorescent markers of protein expression and morphological characteristics such as cell size, it is possible to determine the heritability of biologically-relevant phenotypes. Lineage trees have been used to study a variety of cell systems including developing embryos [6,8–12], bacterial populations [13,14], stem cells [15–17], and lymphocytes [9,18,19].

Our goal is to develop a method to automatically track and segment a population of dividing T cells. T cells provide essential protection against pathogens and cancers in vertebrates and are increasingly used as cellular therapies. T cells can form memories of specific antigens, and so respond more rapidly to subsequent encounters with that antigen. Such memory reflects one of several state changes that can occur during T cell differentiation and activation. Understanding how these state changes, or fates, are controlled is therefore of profound interest and has been the subject of decades of intensive research [20]. A key outstanding question is the extent to which such fate decisions are inherited from one generation to the next, are guided by external influences, or are random. There is clear evidence for all three types of influence, but a quantitative framework is still elusive [21]. The lineage trees that have proven so valuable in mapping heritability in other organisms or cell types have had less impact for T cells (). This is in part because T cells, like many blood cells, are more difficult to track due to their comparatively high motility [22]. However, recently progress in imaging over several generations has provided initial promising findings that indicate strong heritability of fate characteristics such as cell size and proliferative capacity [7,23].

Lineage tracing places stringent requirements on the accuracy of image analysis routines since each cell must be assigned its correct position in the tree [24,25]. A single misidentification error can result in an entire clone of descendants being assigned the wrong common ancestor. This challenge is compounded by the need to keep cells alive over many generations (several days or more), necessitating low intensity illumination (compromising image quality and segmentation accuracy) and low frame rates (compromising tracking accuracy) [2,3]. Under these restrictions, automated tracking of highly motile cells such as T lymphocytes is prone to misidentification error since cells can undergo sizeable changes in position and shape from one image to the next. Previous efforts to develop an automated tracking system for T cell lineage tracing have required extensive manual interventions to correct misidentification errors [24]. A variety of methods have been developed to track multiple cells [2,3,26,27], although few have addressed the specific challenge of tracking highly-motile T cells [24]. Most multi-cell tracking methods adopt the tracking-by-detection paradigm [28–33]. In this approach, segmentation is performed first, on single images, allowing a group of pixels to be detected as an individual cell. Following detection, association is performed, whereby a given cell detected in different images is assigned the same identity [34]. Tracking thus involves detecting in single images and associating between images. It is during the association step that cell misidentification can occur [3].

Deep learning (DL) methods have increasingly shown better performance than classical computer vision methods for microscopy image analysis [27,32]. These DL models have primarily been used for segmentation alone, with association between images being performed by other methods [30,33,35,36]. Increasingly, DL techniques have been used for both detection and association [28,29,31,35]. Nevertheless, these methods all perform association as a separate step after detection making association susceptible to detection errors.

Several approaches using deep learning algorithms have emerged recently which attempt to overcome this limitation in the tracking-by-detection paradigm. In one approach, detection and association are incorporated into a single DL model and performed simultaneously

[37–39]; in another approach, association and detection correct each other during execution [40,41]. Recently it was shown that an effective way to implement feedback between detection and association in a DL model is to use seeds in the previous frame [31,42], a method that was originally used in classical computer vision approaches [43–45]. These efforts all point to new opportunities for improving tracking for lineage analysis.

We have found that the problem of tracking highly-motile T cells with sufficient accuracy to construct a cell lineage tree is not addressed adequately by existing methods. To address this research gap, we have developed a new tool, DeepKymoTracker, that automatically tracks and segments highly-motile T lymphocytes with the accuracy needed to construct a cell lineage tree. We use a 3D convolutional neural network (CNN) for tracking, U-Nets [46] for segmentation, and seeds to integrate detection, association, and segmentation [43–45,47–49]. Conceptually, the 3D CNN implements the idea that the lineage tree can be constructed in space and time (Fig 1A). This is, to the best of our knowledge, the first time a 3D CNN has been used for cell tracking, although a 3D CNN was utilized recently for cell motility prediction [50,51].

The paper is structured as follows. In the Results section, we describe the tracking and segmentation methods used in DeepKymoTracker and the seed mechanism that integrates them. We show results from testing on time-lapse microscopy data from T-lymphocytes and compare them against those from five previously-published cell analysis tools. In the Discussion section we examine the implications of this work. Further details about the algorithm and data are provided in the Materials and Methods section.

## Results

DeepKymoTracker performs tracking and segmentation of cell pedigrees recorded in time-lapse microscopy movies. It accepts a movie as input and generates summary statistics for each cell as output. In addition, an output movie, consisting of the segmented and tracked version of the input movie, is generated for visualisation and validation. An example can be viewed in Fig 1C (the last frame of the output movie) and here (the whole movie).

The DeepKymoTracker pipeline is composed of tracking, segmentation, and division-detection modules, executed in that order (Fig 1B). Tracking is first used to locate the centroid of each cell. Segmentation is then applied to a small patch of the image surrounding that centroid, with the segmented region being used to update the centroid position of the cell. Division detection checks the shape of the segmented region of each cell to determine whether mitosis has occurred and updates the number of cells accordingly. These operations are performed on clips of 4 images at a time.

### Principles of the tracking module

The tracking module in DeepKymoTracker records cell identities and centroid positions from one image to the next. This is accomplished using a 3D convolutional neural network (CNN) and does not involve segmentation (which comes later).

In the 3D CNN, the 2 spatial dimensions (x-y) and 1 temporal dimension (t) are treated similarly, allowing cells to be tracked over multiple frames at once. Within this framework, motion of a cell in x-y-t describes a volume curve (with branches upon cell division, Fig 1A). By including time in the convolutional kernel, the traditional two-part tracking-by-detection paradigm involving detection in separate images followed by association across images becomes a single, self-consistent 3D problem of locating a 'tube'.

In practice, it is not possible to execute the 3D CNN on the whole movie at once (typically thousands of images). Instead, the 3D CNN is executed on short clips, each consisting of 4 consecutive frames. This means that the original challenge of association across adjacent images

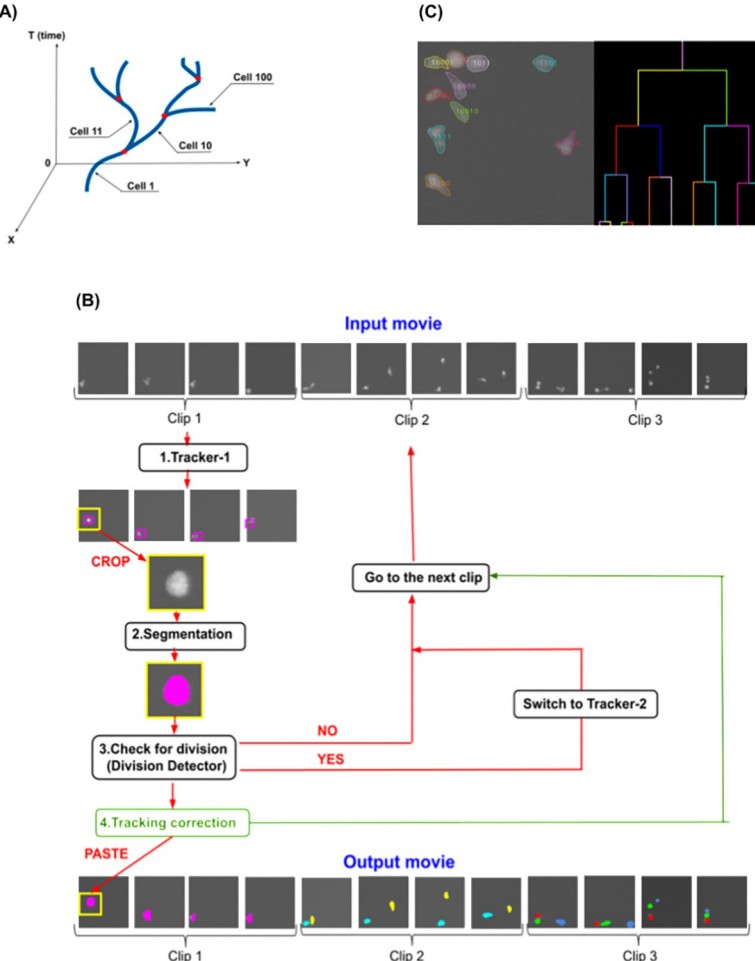

**Fig 1. The main principles of DeepKymoTracker.** (A) Schematic of a lineage tree in 2 spatial and 1 temporal dimension. Tracking involves identifying the position of each of the branches (cell divisions) in the tree and ascribing cell identity based upon relationships between each cell. The output therefore integrates space, time, and lineage information. (B) Overview of the algorithm. The input data consists of a raw movie (top row) divided into clips composed of four frames each. The analysis has 4 basic steps: (1) The tracking module calculates the centroid of each cell in each frame of the clip. (2) Patches of the raw images, centred on each centroid, are fed into the segmentation module, producing segmented versions of each patch. (3) Each segmented patch is then passed to the Division Detector to determine if the cell has divided. If division is detected the algorithm switches to Tracker-2 and the next clip starts with the frame where the division occurred. (4) The centroid of each segmented cell is then used to update the centroid position (green arrow). The entire procedure is then repeated on the next clip. One output is a movie with each cell identified and segmented (bottom row). (C) The output movie of the algorithm: Segmented and tracked cells, along with the lineage tree.

became one of association across adjacent clips. While there are fewer clips than images, the transitions between clips still represent points where cells can more easily be misidentified.

To address this, while remaining within the CNN framework, we introduced a 'seed' channel, or image, to each clip (Fig 2A). This is a blank image to which are added visual markers, or seeds, each slightly larger than the size of a cell (13×13 μm, or 40x40 pixels in our case), located at each cell position. Each seed encodes information about a cell's identity and centroid from the previous clip. By executing the CNN on a seed channel together with the 4 data images in the current clip, the CNN automatically associates cells between clips as well as within clips. This allows the 3D CNN to track cells across multiple clips without the need for a specialized

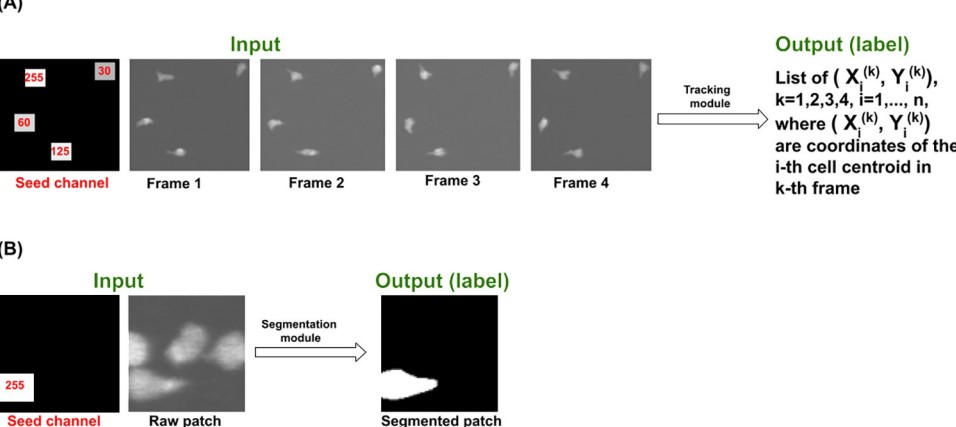

**Fig 2. Examples of seeded inputs to tracking and segmentation modules.** (A) Seed-driven tracking (for Tracker-4): The seed channel is created during execution based on the recalculated positions of the segmented cells in the last frame of the previous clip. The calculated centroids give rise to the seeds, squares of size 40x40 pixels (or the average size of a cell), each with unique intensities (255, 125, 60 and 30) for identification (see Table 1). The seed channel is a simplified version of the last frame of the previous clip. The tracking module uses the seed channel to initialize association in Frame 1; the outputs are the centroids and the identities of the cells inside the clip. (B) Seed-driven segmentation: A seed image is used to identify which of the cells in the patch is to be segmented. The seed channel is created during execution based on the position of the cell in the previous frame.

association module. We found that addition of the seed channel significantly improves association across clips, reducing misidentification errors. This is likely because the seed is a sharp-edged feature that is more readily detected by the CNN than is the image of the original cell. It thus biases the CNN towards preserving cell identities.

**Table 1. Detailed description of the seeds for tracking and segmentation modules.**

| | Tracking module | Segmentation module |
|---|---|---|
| Model | 3D CNN | Ensemble of 2 consecutive 2D U-Nets |
| Goal | Track (i.e., detect and associate) cells in a raw input clip | Segment a tracked cell in a raw patch |
| Type of detection | Centroid calculation | Segmentation |
| Input | A clip of raw movie + seed channel attached to the 1st frame of the clip | Raw patch with a tracked cell + seed channel. |
| Output | Centroids of cells inside the input clip + their identities | Binary patch where the cell of interest is segmented only, others are ignored. |
| Description of seed channel | An artificial frame with squares of *different* intensities. The size of each square is slightly larger than the typical size of a cell. Each is located at the centroid of the cell in the last frame of the previous clip. | An artificial frame (of the same size as the input raw patch) with 1 square of intensity 255. The size of the square is slightly larger than the typical size of a cell. It is located at the centroid of the cell of interest in the last frame of the previous clip. |
| Information contained in the seeds | The locations and identities of the cells in the last frame of the previous clip. | The location of the cell of interest in the previous frame |
| Function of the seeds | Initialize association in the 1st frame of the input clip. | Points to the cell of interest in the input patch telling the neural network which cell in the patch must be segmented. |
| Purpose of the seeds | The seed channel contains information from the segmentation module, and improves tracking, i.e., here segmentation corrects tracking. | The seed channel contains information from the tracking module, and improves segmentation, i.e., here tracking corrects segmentation. |

Each 3D CNN is trained to track a specific number of cells. Thus, for example, Tracker-1 would be used when there was only 1 cell present, Tracker-2 when there were 2 cells, and so on. As we discuss later, these trackers can be combined to track movies with an arbitrary number of cells giving comparable performance to the specialized trackers and only a small reduction in execution speed.

## Principles of the segmentation module

Upon completion of tracking, the clip is passed to the segmentation module. This consists of two consecutive U-Nets. Segmentation returns pixel-level information about each cell as well as summaries such as the size, shape, and brightness of each cell.

To improve accuracy, segmentation is restricted to small patches of the image, each centered on the cell centroids initially estimated by the tracking module. Each patch is approximately 4 times the area of a cell. This size tolerates errors in tracking yet contains only about 6% of the pixels in the full image. Importantly, this eliminates the need to acquire segmentation training data on full images since training can now be performed on small patches alone.

When cells crowd together, multiple cells can appear even within the small patch region, potentially resulting in a misidentification of cells. We thus designed our segmentation module to segment only the cell of interest (designated with a seed) in an input patch (Fig 2B). This seed is very similar to those used in tracking (Table 1).

During execution, the seed channel for segmentation is automatically created based on the position of the cell of interest in the previous frame and represents the visually encoded information about the location of the cell of interest in the previous frame to be passed to the current frame. Before analyzing the current input patch, the segmentation module reads the information in the seed channel of the patch and learns which cell in the patch is meant to be segmented and which cells are to be ignored.

In practice, we observed that the presence of several closely packed cells can cause tracking errors. To correct these, a new centroid is calculated from the segmented cell region. This is then used to reposition the patch and re-segment the cell (Fig 3). It is this updated centroid that is recorded as the final, tracked position and used to locate the seed in the next image. Segmentation is thus used to fine-tune the centroids initially output by the tracking module.

## Benchmark comparison of semantic segmentation, detection, and tracking performance

We benchmarked the performance of DeepKymoTracker on sections of T-cell movies that each represented a different challenge. In Movie 1 (containing 4 cells), the cells had

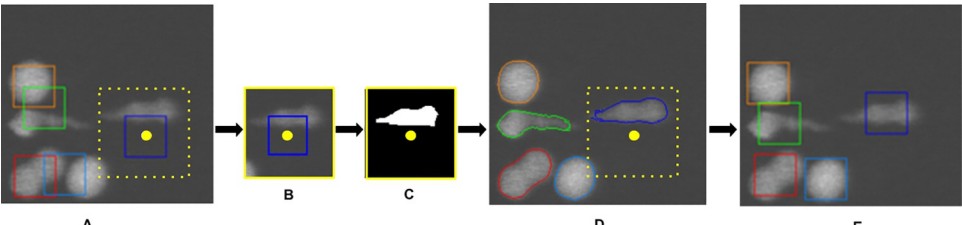

**Fig 3. An example of how segmentation corrects tracking during execution.** (A) Coloured bounding boxes represent where the tracker initially locates the centroids for each of the 5 cells (the yellow dot illustrates the calculated position of a centroid for the case of the blue bounding box). Note how the presence of multiple closely packed cells causes noticeable errors in centroid positions. For each cell, a patch (yellow box) (B) is cropped out, analysed by the segmentation module (C) and the result pasted back (D) into the original image. The centroid positions of each cell are then recalculated from the segmented region. These updated centroid positions (E) now closely track the cells.

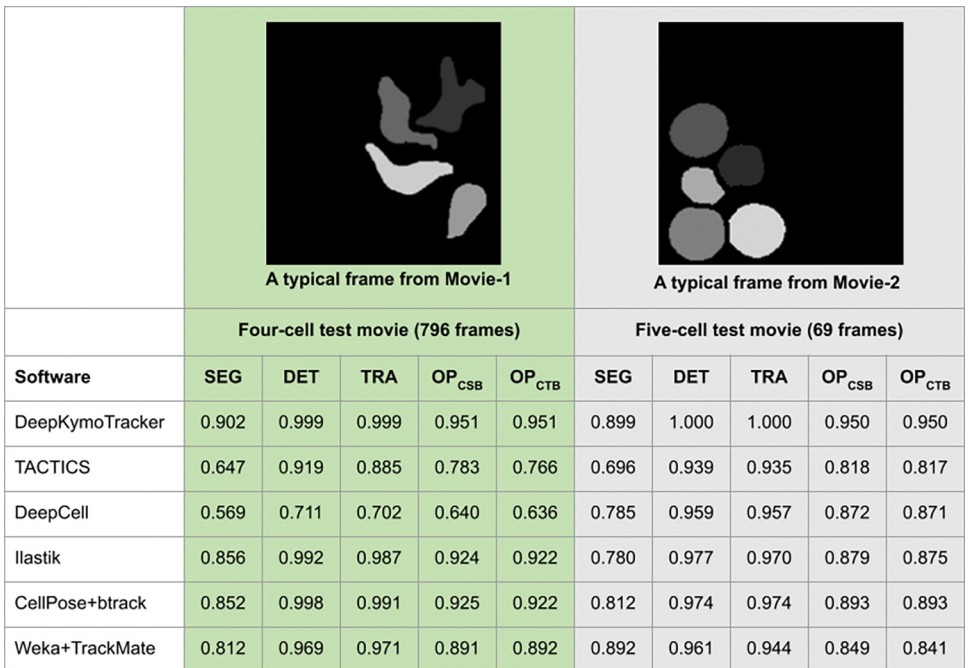

| | Four-cell test movie (796 frames) | | | | | Five-cell test movie (69 frames) | | | | |
|---|---|---|---|---|---|---|---|---|---|---|
| Software | SEG | DET | TRA | OP$_{CSB}$ | OP$_{CTB}$ | SEG | DET | TRA | OP$_{CSB}$ | OP$_{CTB}$ |
| DeepKymoTracker | 0.902 | 0.999 | 0.999 | 0.951 | 0.951 | 0.899 | 1.000 | 1.000 | 0.950 | 0.950 |
| TACTICS | 0.647 | 0.919 | 0.885 | 0.783 | 0.766 | 0.696 | 0.939 | 0.935 | 0.818 | 0.817 |
| DeepCell | 0.569 | 0.711 | 0.702 | 0.640 | 0.636 | 0.785 | 0.959 | 0.957 | 0.872 | 0.871 |
| Ilastik | 0.856 | 0.992 | 0.987 | 0.924 | 0.922 | 0.780 | 0.977 | 0.970 | 0.879 | 0.875 |
| CellPose+btrack | 0.852 | 0.998 | 0.991 | 0.925 | 0.922 | 0.812 | 0.974 | 0.974 | 0.893 | 0.893 |
| Weka+TrackMate | 0.812 | 0.969 | 0.971 | 0.891 | 0.892 | 0.892 | 0.961 | 0.944 | 0.849 | 0.841 |

**Fig 4. The cell benchmark evaluation results.** This figure shows the performance of different cell analysis tools on T-cell movies with either complex shapes (left) or close packing (right). DeepKymoTracker outperformed other methods in all aspects: SEG (semantic segmentation measure), DET (detection measure) and TRA (tracking measure). DeepKymoTracker had the highest Cell Segmentation Benchmark ranking OP$_{CSB}$ and Cell Tracking Benchmark ranking OP$_{CTB}$.

complicated shapes while in Movie 2 (containing 5 cells) the cells, although more round, were often closely packed (see images in Fig 4). In these tests, dedicated trackers were used. Thus, Tracker-4 was used for the 4-cell movies and Tracker-5 for the 5-cell movies. Later we describe how combining smaller trackers (such as using Tracker-1 four times instead of Tracker-4 once) does not appreciably degrade performance.

We compared DeepKymoTracker to several publicly available tools previously demonstrated in cell biology applications. Each method was selected because at least one of its components, segmentation or tracking, was based on machine learning or deep learning approaches. The methods chosen were DeepCell [52,53], Ilastik [54], Weka [55] for segmentation plus TrackMate [56,57] for tracking, CellPose [58] (model Cyto-2) for segmentation and btrack [59] for tracking. In addition, we included TACTICS [24,25], which is built upon traditional computer vision algorithms and was used to semi-automatically prepare training data for our neural networks.

To compare performance of these algorithms, we used the Cell Tracking Challenge [60] evaluation methodology, a popular tool adopted by the cell imaging community. The Cell Tracking Challenge involves both tracking and segmentation tasks. Results are supplied in terms of 3 measures: semantic segmentation measure (SEG), detection measure (DET), and tracking measure (TRA). Methods can be ranked by the Cell Segmentation Benchmark (OP$_{CSB}$), which is the average of DET and SEG measures, and the Cell Tracking Benchmark (OP$_{CTB}$), which is the average of DET and TRA measures.

The performance measures for each algorithm are shown in Fig 5. DeepKymoTracker outperforms other methods in all three aspects of cell analysis: semantic segmentation, detection, and tracking.

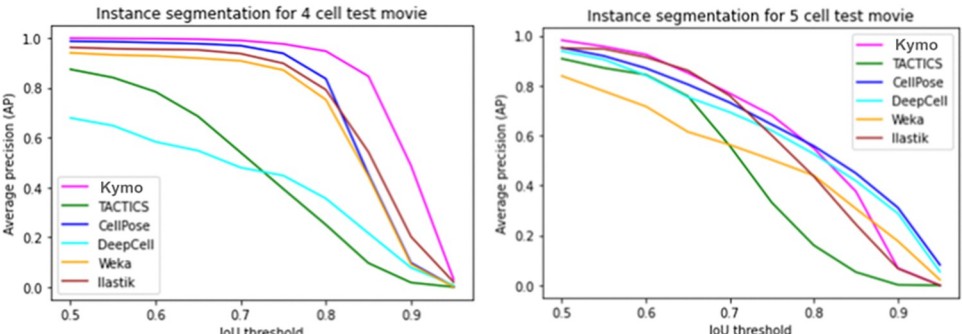

**Fig 5. Average precision curves for instance segmentation at various IoU thresholds.** Left: AP for the 4-cell test movie (796 frames). DeepKymoTracker shows superior performance throughout. Right: AP for the 5-cell test movie (69 frames). DeepKymoTracker shows superior performance up to IoU = 0.8 but degrades more rapidly than some other algorithms at higher IoU (this is likely due to the shortage of training data for 5 cells).

In the 4-cell test movie, DeepKymoTracker was the top performer in all metrics (Table 2) and had the highest AP at all IoU thresholds (Fig 5, left). DeepKymoTracker thus appears to have had little difficulty with complex cell shapes.

## Benchmark comparison of instance segmentation

The Cell Tracking Challenge provided useful metrics for overall performance. Here we explore other metrics used in classification problems that provide further insight into segmentation performance.

We examine the detection true positive rate (TP), false positive rate (FP), false negative rate (FN), precision (Precision), recall (Recall) and F1-score. In addition, we report the average

**Table 2. Instance segmentation benchmarking results.**

| Software | SEG | $TP_{0.5}$ | $FP_{0.5}$ | $FN_{0.5}$ | $Precision_{0.5}$ | $Recall_{0.5}$ | $F1\text{-Score}_{0.5}$ | $AP_{0.5}$ | $AP_{0.75}$ | avAP |
|---|---|---|---|---|---|---|---|---|---|---|
| **Four-cell test movie (796 frames)** | | | | | | | | | | |
| **DeepKymoTracker** | 0.902 | 0.999 | 0.001 | 0.001 | 0.999 | 0.999 | 0.999 | 0.999 | 0.976 | 0.827 |
| **TACTICS** | 0.647 | 0.898 | 0.061 | 0.102 | 0.922 | 0.898 | 0.908 | 0.874 | 0.394 | 0.448 |
| **DeepCell** | 0.569 | 0.719 | 0.121 | 0.282 | 0.849 | 0.719 | 0.765 | 0.729 | 0.448 | 0.208 |
| **Ilastik** | 0.856 | 0.979 | 0.027 | 0.021 | 0.973 | 0.979 | 0.975 | 0.962 | 0.898 | 0.722 |
| **CellPose** | 0.852 | 0.998 | 0.014 | 0.002 | 0.988 | 0.998 | 0.992 | 0.987 | 0.938 | 0.722 |
| **Weka** | 0.812 | 0.953 | 0.027 | 0.047 | 0.968 | 0.953 | 0.959 | 0.940 | 0.871 | 0.679 |
| **Five-cell test movie (69 frames)** | | | | | | | | | | |
| **DeepKymoTracker** | 0.899 | 0.988 | 0.012 | 0.012 | 0.988 | 0.988 | 0.988 | 0.982 | 0.681 | 0.616 |
| **TACTICS** | 0.696 | 0.942 | 0.061 | 0.058 | 0.941 | 0.942 | 0.942 | 0.908 | 0.331 | 0.449 |
| **DeepCell** | 0.785 | 0.948 | 0.0016 | 0.054 | 0.985 | 0.998 | 0.964 | 0.938 | 0.620 | 0.604 |
| **Ilastik** | 0.780 | 0.965 | 0.026 | 0.035 | 0.973 | 0.965 | 0.968 | 0.951 | 0.600 | 0.578 |
| **CellPose** | 0.812 | 0.968 | 0.032 | 0.032 | 0.969 | 0.968 | 0.969 | 0.952 | 0.644 | 0.632 |
| **Weka** | 0.737 | 0.878 | 0.075 | 0.122 | 0.913 | 0.878 | 0.894 | 0.839 | 0.503 | 0.496 |

Shown are the true positive rate ($TP_{0.5}$), false positive rate ($FP_{0.5}$), false negative rate ($FN_{0.5}$), precision ($Precision_{0.5}$), recall ($Recall_{0.5}$) and F1-score ($F1\text{-Score}_{0.5}$)–all calculated at a threshold IoU = 0.5. The average precision at IoU = 0.5 ($AP_{0.5}$) and IoU = 0.75 ($AP_{0.75}$) are provided, along with the mean average precision (avAP). For reference, the SEG measure from Fig 5 is also provided.

precision ($AP_t$) at various IoU thresholds $t$,

$$\mathrm{AP_t} = \frac{TP(t)}{TP(t) + FP(t) + FN(t)}, \tag{1}$$

where TP(t), FP(t) and FN(t) are the true positive, false positive, and false negative rates evaluated at a given intersection-over-union (IoU) threshold $t$. We also report the mean average precision (avAP), defined as

$$avAP = \frac{1}{10}\sum_{i=0}^{9} \frac{TP(t)}{TP(t) + FP(t) + FN(t)}, \ t = 0.05i + 0.5, \tag{2}$$

calculated at 10 IoU thresholds between 0.5≤t ≤0.95, in step sizes of 0.05. Results are summarized in Table 2. Graphs showing AP at different IoU thresholds have been plotted and displayed in Fig 5.

On the 5-cell test movie DeepKymoTracker was the top performer in almost all metrics (Table 2). The single exception was in the mean average precision avAP = 0.632, where Cell-Pose outperformed DeepKymoTracker (avAP = 0.616) slightly. Fig 5 (right) shows how this underperformance arises only at IoU thresholds above 0.8. This may have been due to a shortage of training data for 5 cells. It should be noted that the execution time for CellPose was about 10 times slower than for DeepKymoTracker.

Note that false positive rates $FP_{0.5}$ and false negative rates $FN_{0.5}$ were much lower for DeepKymoTracker, which uses a pre-determined number of detections. Thus, the number of objects (cells) in each frame is equal to the number of predictions.

## Comparison of performance using generalized versus specialized trackers

In the benchmark tests described above, which were performed on movies with a fixed number of cells, we used a specialized CNN that was trained on exactly the number of cells in the movie. We had hypothesized that using such specialized trackers would achieve the highest tracking accuracies and many of the movies of T cells involve only a few cells. This approach, however, becomes impractical with more cells since training data would be required for every possible number of cells.

A more general approach is to combine trackers, each of which is trained on fewer cells provided they learned to ignore other cells during training. We call such trackers general ones, i.e. instead of term Tracker-2, for instance, we use Tracker-2-general. Thus, for 5 cells, one could apply Tracker-2-general for the first two cells and Tracker-3-general for the next 3. We find that, *by using a seed image during both training and execution, our generalized tracking CNN ignores cells that are not seeded*. Thus, Tracker-*n*-general, which is trained on movies with more than $n$ cells where only n cells are seeded, can be applied to movies with >$n$ cells provided only $n$ cells are seeded. The remaining $m$ cells can be tracked with a Tracker-*m*-general. This highlights how the seed image decomposes the multi-object tracking problem into a sum of simpler tracking problems. We experimented with different combinations of general trackers and observed that any number of cells can be tracked with Tracker-1-general where only one cell in the frame is seeded.

Surprisingly, there was essentially no loss in accuracy when going from specialized trackers to the generalized tracker Tracker-1-general. We tested Tracker-1-general on the movies shown in Fig 4. Thus, for the 5-cell movie we compared the performance of Tracker-5 with that of Tracker-1-general alone. For the 4-cell movie we compared the performance of Tracker-4 with that of Tracker-1-general alone. Similar comparisons were made for 2- and 3-cell movies. As can be seen in Table 3, Tracker-1-general shows very similar SEG, DET, and TRA measures to the specialized trackers.

**Table 3. Comparison of specialized versus generalized tracker Tracker-1-general.**

| Tracker 5 | | | 5 x Tracker-1-general | | |
|---|---|---|---|---|---|
| SEG | DET | TRA | SEG | DET | TRA |
| 0.899 | 1.000 | 1.000 | 0.898 | 1.000 | 1.000 |
| Tracker 4 | | | 4 x Tracker-1-general | | |
| SEG | DET | TRA | SEG | DET | TRA |
| 0.902 | 0.999 | 0.999 | 0.894 | 0.999 | 0.999 |

There was little, if any, difference in segmentation (SEG), detection (DET) and tracking (TRA) accuracy.

In addition to assessing accuracy above, we compared execution times (Table 4). We found that applying Tracker-1-general multiple times is generally slower than applying a single tracker (such as Tracker-5) to do the same task. If extensive tracking is to be performed on large movies with similar numbers of cells, it thus makes sense to train dedicated trackers. Otherwise, the generalized tracker is likely adequate for most applications.

## Discussion

The task of automatic measurement of cell lineage trees using time-lapse microscopy places severe requirements on the accuracy of multi-object tracking algorithms. Since just a single identity swap can affect inferences about trait heritability and fate determination, each cell's identity must be tracked and preserved across hundreds or thousands of images. This is particularly challenging for highly motile cells such as T lymphocytes. To address this, we developed DeepKymoTracker, an automated pipeline for tracking and segmenting proliferating cells. DeepKymoTracker was benchmarked against five existing software tools for automated cell tracking, outperforming them in almost all aspects of segmentation, detection, and tracking.

**Table 4. Comparison of execution times for different combinations of trackers on different hardware.**

| | RAM = 16 GB (Laptop) | RAM = 8 GB (Desktop) | RAM = 128 GB (Lambda machine without GPU) | RAM = 128 GB (Lambda machine with 1 GPU) |
|---|---|---|---|---|
| **2 cells (1120 frames)** | | | | |
| **2 x Tracker-1-general** | 1.667 | 0.725 | 0.544 | 0.154 |
| **Tracker-2** | 1.533 (**8%**) | 0.654 (**9.8%**) | 0.517 (**5%**) | 0.152 (**1.2%**) |
| **3 cells (128 frames)** | | | | |
| **3 x Tracker-1-general** | 2.400 | 1.140 | 0.759 | 0.203 |
| **Tracker-3** | 2.133 (**11%**) | 0.967 (**15.2%**) | 0.728 (**4.1%**) | 0.195 (**3.9%**) |
| **4 cells (910 frames)** | | | | |
| **4 x Tracker-1-general** | 3.250 | 1.467 | 1.017 | 0.234 |
| **Tracker-4** | 2.750 (**15.4%**) | 1.233 (**16%**) | 0.907 (**10.8%**) | 0.234 (**0%**) |
| **5 cells (69 frames)** | | | | |
| **5 x Tracker-1-general** | 3.855 | 1.788 | 1.282 | 0.319 |
| **Tracker-5** | 3.188 (**17.3%**) | 1.424 (**20.4%**) | 1.166 (**9.1%**) | 0.304 (**4.6%**) |

Shown are execution times, in seconds per frame, for different tracker combinations, applied to movies containing 2–5 cells. The numbers in parentheses show the percentage reduction in execution time compared to the case where only Tracker-1-general is used. Using Tracker-1-general is generally slower than using the larger trackers. The difference is negligible in higher-performance GPU-based machines.

DeepKymoTracker uses a 3D convolutional neural network (CNN) that integrates detection and association into one step, reducing errors that arise in the commonly used two-step tracking-by-detection framework. By tracking before segmentation, the latter can be performed on small image patches containing each cell, achieving high accuracy without having to segment unnecessary regions of the image. We provide a way for the high-resolution segmentation result to update the tracking result, significantly improving tracking accuracy in crowded images. This scheme leverages the prior knowledge that, in these types of cell cultures, cells constitute only a few percent of the image area. Thus better performance can be achieved by focusing primarily on the parts of the image containing cells.

Seed images play a central role in preserving cell identities. They transmit cell identity information between movie clips and between tracking and segmentation modules, reducing the chance of identity swaps during transitions between analysis steps.

We observed that seeds can be used to identify the subset of the cells which are to be tracked and segmented. Cells which are not labelled with a seed are simply ignored in the analysis. This is a useful practical result because it means that networks trained on movies of a few cells can be applied to data with many cells, with no reduction in accuracy. This result has broader implications. It suggests that seeds combined with deep neural networks decompose the multi-object tracking problems into a sum of single-object tracking problems. It will be important to explore whether this idea generalizes beyond the examples used here. Although we have demonstrated superior performance of DeepKymoTracker in segmentation, detection, and tracking, our testing was only performed on cells in the first few generations of a lineage tree, since this has been the focus of our current applications. As we explore data in the later stages of a tree, the effects of over-crowding and occlusions on tracking accuracy will become significant. We anticipate that the techniques we have described here, using seed-based 3D convolutional networks for multi-object tracking, can be generalized to tracking in crowded environments.

In this study our focus was on eliminating the identity errors that occur when tracking and segmenting of T cells. Because there were only a few cells moving inside an essentially 2D suspension, occlusions were relatively rare. In addition, in the early part of the lineage tree studied here there was little cell death and we could assume that cell numbers stayed the same or increased (by division). As we move to the study of larger trees, both occlusions and cell death become significant and improvements will need to be made to DeepKymoTracker. These improvements will involve a more sophisticated running count of cells than is currently used in the division tracker.

The usefulness of DeepKymoTracker for studying other types of cells depends on the type of data involved. Our tool was trained on 2D microscopy images that included both bright field and fluorescent image channels (see Materials and Methods), involved only a few cells per image, and had few occlusions. Assuming that these conditions are satisfied, we expect DeepKymoTracker to perform well without additional training. Preliminary tests have verified that this is indeed the case for movies of developing T cells ('DN3 cells'). We expect that, because the mature T cells used in training adopt a variety of shapes and exhibit erratic movement patterns, the method is versatile enough to perform well on cells with different motility patterns and morphological characteristics.

## Materials and methods

### Image acquisition and software implementation

1. **T-cell data.** T cells from OT-1 transgenic mice were incubated for 40 hours with peptide-pulsed Dendritic Cells taken from mice bone marrow. Mouse T cells in the time lapse images were obtained under Peter MacCallum Cancer Centre Animal Ethics Committee

approved protocol E535. Cells were tagged with Green Fluorescent Protein to enhance tracking [61].

2. **Instrumentation.** Microscopy images of T cells were obtained with an IX71 inverted microscope (Olympus, Tokyo, Japan) and EM-CCD Andor camera (Model: iXon EM +885, Belfast, Northern Ireland) [25]. Multiple stage positions were captured with varying sampling rate (e.g., 30 sec, 1 min, 2 min or 10 min) for several days and are stored in the form of movies of roughly 3,500 frames. The microscope magnification was typically 20×, giving 0.327 μm/pix. 2 channels for each frame, fluorescent and bright field, were used in this project. The size of the well in each image is 125 x 125 μm$^2$ (382 x 382 pixels).

3. **Training data** for the neural networks was obtained using TACTICS [25], a MATLAB-based cell tracking and segmentation software developed at Swinburne University of Technology. TACTICS follows the conventional order, segmenting cells before tracking. Segmentation is performed using traditional computer vision techniques. Thus, the user performs manual segmentation of one or two images using the TACTICS interface and the resulting parameter settings are applied to the rest of the movie. The cases of cells occluding and touching each other present a particular challenge for TACTICS and usually require manual correction. The Hungarian tracking algorithm [62] is used to associate cells in segmented images. This often requires manual post-correction.

4. **Software implementation.** The code was written in Python 3.6.4., Keras 2.1.6. and TensorflowGPU 1.8.0. Training of neural networks was performed on OzStar, a Swinburne University Supercomputer, where two NVIDIA P100 12GB PCIe GPUs were utilized for each training session. TACTICS software and the OpenCV library was deployed for creating training and test data for both segmentation and tracking neural networks. At the execution stage, ImageJ 1.53q and OpenCV were used to create output videos with segmented and tracked cells.

## Tracking module

DeepKymoTracker performs tracking prior to segmentation. The main features of the tracking module are as follows:

**1. Tracking n cells.** Several convolutional neural networks for tracking were trained—namely, Tracker-1, Tracker-2, Tracker-3, Tracker-4 and Tracker-5 each specializing in tracking a fixed number of cells., i.e., Tracker-1 was trained to track one cell only, Tracker-2 –two cells, and so forth. At execution, the algorithm switches from Tracker-N to Tracker-N+1 when a division of a cell occurs which is detected by the Division Detector. This means that different trackers are employed for different sections of the lineage tree.

**2. Only fluorescent segmented images were utilised for training.** This reduced the effort required to produce segmented images for training our tracking neural networks. Note that bright field channel was used for training the segmentation module (see below).

**3. Synthetic data.** Due to the scarcity of real data, synthetic images were used to train Tracker-3, Tracker-4, and Tracker-5. Simulating artificial cells required modelling 3 features: 1. The starting positions of cells in a clip. 2.The shapes of artificial cells. 3. Motion of cells within a clip.

The little data that was available was often biased towards certain cell coordinates: thus, only a fraction of all possible positions in a 382 x 382 image were covered (see Fig 6B left which shows the scatter plot of centroid coordinates of one randomly chosen cell from a real

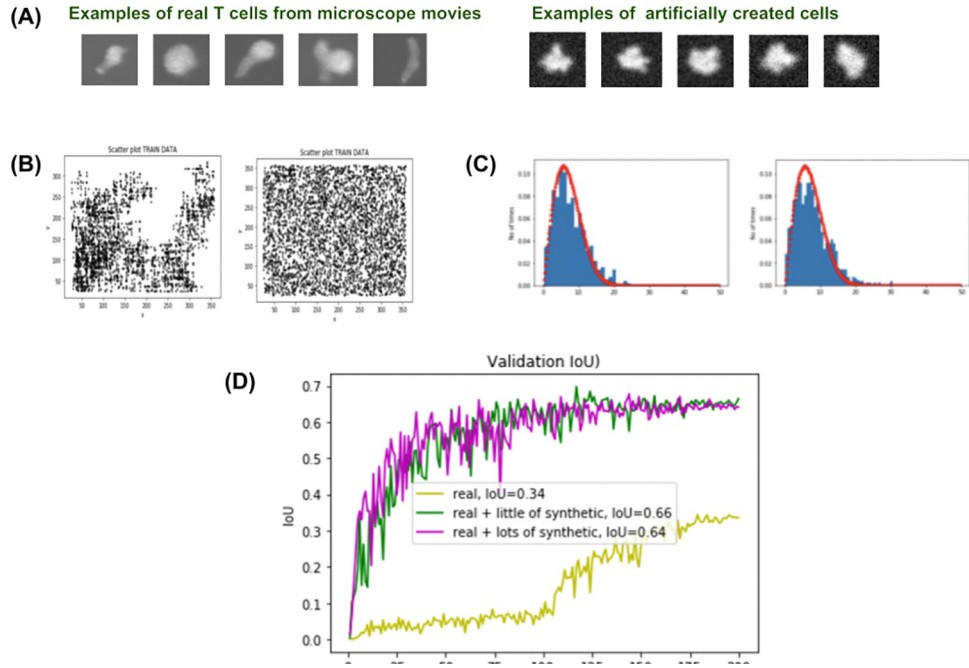

**Fig 6. Creation of synthetic data for the tracking module.** (A) Modelling cell shapes: Examples of real and synthetic T cell shapes. (B) Scatter plots for locations of a real and a synthetic cell within a 382 x 382 frame. Left: A typical scatter plot of a real T cell positions during its lifetime taken from a cell movie. The cell shows a tendency to gravitate towards one side of the image for reasons unknown. Right: A scatter plot of the centroid coordinates of one artificially-generated cell. All the accessible locations within the frame are covered. (C) Histograms of step sizes of T cell motions. These are examples of histograms of step sizes for 2 different T cells. The red line is the fitted distribution function that was used to generate the synthetic data (formula (3). (D) Performance with different percentages of synthetic samples in the training data set.

movie). To remedy this, the coordinates ($x_c$, $y_c$) of artificial cells in the first images of each training clip were generated randomly from uniform distribution in the range (0; 382) (Fig 6B right).

Simulating cell shape involved subjecting the binary image of a filled circle to dilation, the addition of a random interior to the cell, blurring, and the addition of random background. The result is shown in Fig 6A (right). To simulate motion of a cell within a clip, two parameters were chosen: the step size R (measured in pixels) and the angle A (measured in radians) at which the cell moves from the current position to the next one. The angle A was sampled from the random uniform distribution in the range (0, $2\pi$). To model R, the motion patterns of real T cells in microscopic movies were analysed (see Fig 6C). As can be concluded from those histograms, the distribution of the step size follows quite an explicit pattern which is approximated by

$$f(t) = a \cdot \left(\frac{t}{b}\right) \cdot e^{-\left(\frac{t}{b}\right)^2}, \quad t \in [0; +\infty) \tag{3}$$

where $a = 0.25$, $b = 7$. The step size was sampled from this distribution.

To understand how well the created artificial clips approximate the real data, we experimented with Tracker-2; we trained it first on real data and then retrained it by adding more synthetic data. The result is shown in Fig 6D.

It can be seen that there is a point of saturation beyond which the performance of the neural network cannot be improved regardless of how much more artificial data is added.

**Table 5. Training details of the trackers.**

| | Tracker-1 | Tracker-2 | Tracker-3 | Tracker-4 | Tracker-5 |
|---|---|---|---|---|---|
| Number of training clips | 22,464 | 26,592 | 24,480 | 32,868 | 29,984 |
| Number of validation clips | 2,944 | 4,672 | 5,888 | 7,584 | 384 |
| Percentage of real data among training clips | 100% | 100% | 4% | 6% | 0.19% |
| Number of epochs | 150 | 150 | 150 | 150 | 150 |
| Optimizer | Adam | Adam | Adam | Adam | Adam |
| Initial learning rate | 0.009 | 0.01 | 0.01 | 0.01 | 0.01 |
| Batch size | 32 | 32 | 32 | 32 | 32 |
| Training loss | 0.00007 | 0.00010 | 0.00035 | 0.00060 | 0.00079 |
| Validation loss | 0.00012 | 0.00015 | 0.00057 | 0.00071 | 0.00110 |
| Training IoU | 0.891 | 0.81861 | 0.75059 | 0.70954 | 0.69866 |
| Validation IoU | 0.883 | 0.81764 | 0.74573 | 0.70272 | 0.69540 |

Based on these results, the training sets for Tracker-3, Tracker-4 and Tracker-5 were built in the similar fashion: a random number of artificial samples were created and added to the real data as the first approximation and then, after training the neural network on this data, more synthetic data was added little by little until the performance ceased to show any signs of improvement.

This diagram shows validation IoU curves for Tracker-2 after training it on different proportions of synthetic data in the training set. The yellow graph is the validation curve when Tracker-2 was trained on the insufficient amount of real data (1,000 samples). The validation IoU achieved was 34%. The green graph is the validation curve after a moderate amount (26,592) of synthetic data was added to the training set. As can be seen from the plot, the performance was boosted to 66%. The magenta curve was obtained when the synthetic component in the training data had been increased significantly (53,184 samples), which did not lead to any improvements. Thus using synthetic data (~26 times the number of real images) can significantly improve training performance, up to a point.

Despite the simple model for synthetic data it proved to be effective: as can be seen from Table 5, even though the percentage of real images in the training data for the trackers was in the range 0.019% to 6%, the validation performance of the trackers was quite high.

4. **Inputs to the trackers: short sequences of frames**. The tracking neural networks take advantage of 3D convolutions where inputs to the neural networks are clips of cell movies rather than frames, i.e., 3D images where the $3^{rd}$ dimension is time. After exploring CNNs trained on 4, 8, 16, 32 and 64-frame clips it was decided to use 4-frame clips to reduce training time and memory needed.

5. **Regression task**. Tracking was cast as a regression task where the outputs were the coordinates of cells in each frame of the input clip. Thus, the output vector has length 2 x 4 x N where N is the number of cells in each of the 4 frames of the input clip and 2 is the number of coordinates (which are x and y -components of the cell centroid).

6. **The loss function and performance metrics.** The mean squared error (MSE) was used as a loss function and intersection over union (IoU) as a performance metric for the quality of our trackers during training. Our tracking algorithms output centroids of the cells present in the frames of an input clip. "Proxy" bounding boxes were created to calculate the IoU. These were 40 x 40, chosen since cells are approximately 40 pixels in diameter. This

approach was inspired by [35] where the authors utilised similar "proxy" boxes to assess the performance of their multi-cell centroid detection algorithm.

7. **Architecture: four separate output branches.** As the first attempt, a classical pattern "convolution followed by pooling" repeated a few times and finalized by a fully connected output layer was utilized (this is the architecture of AlexNet [63] which was implemented here using 3D instead of 2D convolutions, with some minor modifications in the number of layers and the number of neurons in each layer). The fully connected output layer consisted of 2 x N x 4 neurons where N was the fixed number of cells to be tracked, 4 is the number of frames in each clip, and 2 is the number of coordinates for each cell. However, even though the neural network did learn to output centroids in each frame of the input clip, it tended to average all those coordinates frame wise: as a result, there was no motion of bounding boxes inside a clip, and there were sharp jumps during transition from one clip to another during execution. To remedy this, a multiple output architecture was introduced–instead of one fully connected final layer with one output vector containing coordinates of all the cells in the whole clip, 4 separate branches were implemented, one for each frame (Fig 7A). This improved performance, allowing for motion within each clip, leading to a validation loss decreasing from 0.0076 to 0.0047 and an increase in validation IoU from 0.38 to 0.76 (for Tracker-2).

8. **Architecture: pooling kernels.** Another important characteristic of these neural networks (Tracker-1, Tracker-2, etc.) is that pooling kernels used in pooling layers were all of size (2,2,1): here the third dimension being equal to 1 is essential. This approach was chosen to ensure that there was no pooling along the $3^{rd}$ dimension. Our goal is to ensure that the centroid coordinates in each frame are preserved as accurately as possible.

9. **Architecture: convolutional part and top layers.** The neural networks Tracker-1, Tracker-2, Tracker-3, Tracker-4 and Tracker-5 have a similar architecture (see Fig 7A).

ReLU activations were used in each convolutional layer. We used classical ReLUs, which showed better performance than Leaky ReLU and PReLUs. The convolutional architecture is given in Table 6. Each branch in the top layers consists of 1 or 2 consecutive fully connected layers (the number of these layers necessary for each tracker was established experimentally) plus an output layer.

The number of neurons in these layers is different for different trackers and is shown in Table 7. ReLU activations are utilized in these layers. The output layer consists of 2 x n neurons for Tracker-n where n is the number of cells to be tracked. No activation function is applied to the output layer since it involves a regression task. These neural networks have 4 separate outputs each containing 2 x n neurons representing the coordinates of n cells present in each frame of the input clip. In Fig 7A, they are visualised as bounding boxes of different colours with the centre in the calculated centroid.

10. **Seed channel in the input clip.** Artificially created frames with squares of different intensities (seeds) were attached to the beginning of the input clips to help initialize identities of each tracked cell inside the clip. The positions of artificial squares are positions of the cells in the last frame of the previous clip. The seed image is thus a simplified version of a frame with cells substituted by squares. The identity of each cell in the clip is stored as the pointing square in the seed image which has the largest overlap with the cell. We found this way of assigning identities to be effective in guiding trackers through the learning process. Fig 7B illustrates how much easier it was for Tracker-2 to learn the centroids of the cells when it

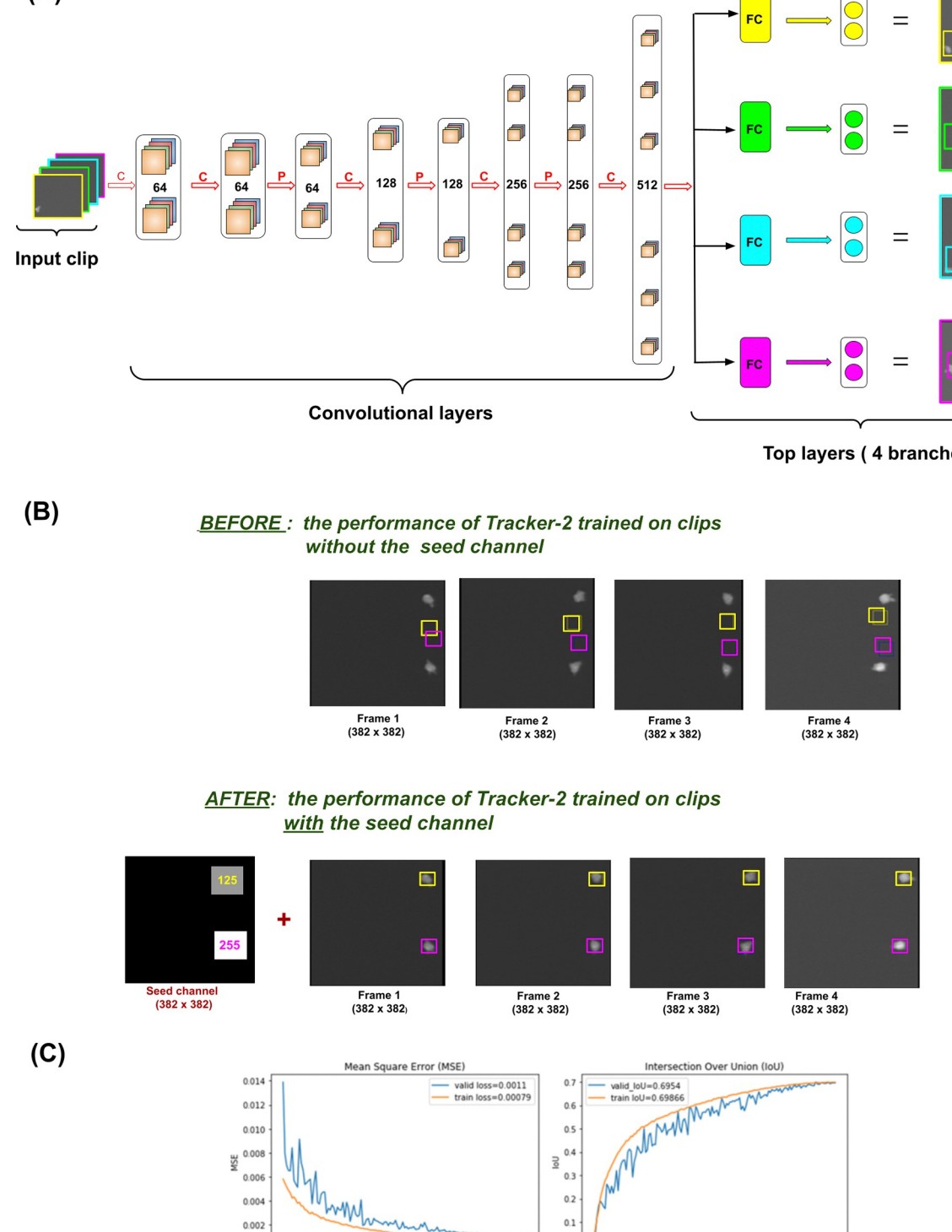

**Fig 7. Tracking module training details.** (A) The architecture of Tracker-1, a neural network for tracking one cell in sections of a cell movie containing one cell only. The inputs are clips of a cell movie comprising 4 consecutive frames of that movie. The convolutional layers consist of a series of convolution (C) and pooling (P) layers with several filters given in the picture. The convolutional part remained unmodified for any Tracker-N, where N = 1,2,3,4,5. The top layers comprise 4 parallel branches each corresponding to its own frame in the input clip: The 1st branch (yellow) corresponds to Frame-1, the 2nd one (green) corresponds to Frame-2, and so forth. Each branch consists of 2 consecutive

layers: 1. Fully connected (FC). The number of neurons in this layer differs for different trackers. Also, one more FC layer following this one is added to each branch for Tracker-2, . . ., Tracker-5. 2. Output layer consisting of 2xN neurons for Tracker-N where N is the number of cells to be tracked. (B) The impact of introducing the seed channel on the performance of Tracker-2. BEFORE: Tracker-2 had difficulty calculating the coordinates of the two cells with close or equal x-coordinates when trained on clips without an artificial frame. AFTER: Implementing the artificial frame with squares of different intensities (255 and 125) each pointing to its own cells resolved the difficulty. (C) The training and validation curves for Tracker-5. Visualization of the training process for Tracker-5: The training and validation curves for mean square error (left) and Intersection Over Union (right).

was provided with the seed channel compared to the method of assigning identities where the cells were distinguished by the x-coordinates of their centroids: the cell with a smaller x-coordinate was assigned identity "Cell 1" and the other was "Cell 2". In Fig 7B (top row), an example of a clip with two cells with very close x-coordinates is given. Tracker-2 struggled to determine which of the two coordinates was smaller. The implementation of the seed channel, however, resolved this problem (Fig 7B, bottom row). With the seed channel the validation loss decreased from 0.0047 to 0.00015 and IoU increased from 0.76 to 0.82 (for Tracker-2).

11. **Resizing and normalization.** All the input frames were resized from 382 x 382 to 100 x 100. Each frame in the input clip was normalized by subtracting the mean of the image from every pixel intensity and then dividing by the standard deviation. Also, the labels which are the coordinates of cells' centroids in each of a 382 x 382 input clip, were divided by 382 to bring them into the range between 0 and 1.

12. **Training procedure.** The training details for each tracker are given in Table 6. As can be seen, there was a shortage of real training data for Tracker-3, Tracker-4 and Tracker-5 which necessitated creation of the synthetic clips: the shapes of T cells and motion patterns were modelled based on their real parameters. The best results were achieved on the mixed training data. The amount of synthetic training data was increased until the neural network performance reached saturation where further addition of the synthetic training samples ceased to improve the performance. An example of the learning curves for Tracker-5 is given in Fig 7C.

13. **Tracker-1-general.** To use Tracker-1 in clips with more than 1 cell, we needed to train a seeded version of Tracker-1 that we named Tracker-1-general. The Tracker-1-general needed to learn to track only the cell of interest (which was identified with a seed) while ignoring all other cells present in a frame. New training data was created for this task. The deep learning model was significantly increased compared to all previous trackers: the

Table 6. **The architecture of the convolutional part of the five trackers.**

| Number | The type of layer | Kernel size | Number of filters |
|---|---|---|---|
| 1. | Convolutional | 3 x 3 x 3 | 64 |
| 2. | Convolutional | 3 x 3 x 3 | 64 |
| 3. | Pooling | 2 x 2 x 1 | 64 |
| 4. | Convolutional | 3 x 3 x 3 | 128 |
| 5. | Pooling | 2 x 2 x 1 | 128 |
| 6. | Convolutional | 3 x 3 x 3 | 256 |
| 7. | Pooling | 2 x 2 x 1 | 256 |
| 8. | Convolutional | 3 x 3 x 3 | 512 |

The convolutional part of the neural network was the same for trackers 1 through 5.

**Table 7. The architecture of each branch in the top layers of the trackers.**

| Tracker | FC–1 layer (# of neurons) | Dropout layer | FC-2 layer (# of neurons) | Output layer (# of neurons) |
|---------|---------------------------|---------------|---------------------------|------------------------------|
| Tracker-1 | 256 | - | - | 2 |
| Tracker-2 | 960 | - | 512 | 4 |
| Tracker-3 | 1024 | 0.05 | 512 | 6 |
| Tracker-4 | 1024 | 0.03 | 512 | 8 |
| Tracker-5 | 1024 | 0.05 | 512 | 10 |

The top layers of the trackers consisted of 4 parallel branches (according to the number of frames in each clip) with the same architecture within each tracker. In this table, the details of each branch are provided. Fully connected layer FC-1 is followed by fully connected layer FC-2 for all trackers except Tracker-1 (which has only FC-1 layer in each branch). Dropout is applied to some of these layers.

convolutional layers with 64 and 128 filters were doubled and one more double layer with 32 filters was added in the front of the previous convolutional part (given in Table 6). The number of neurons in layer FC-1 for Tracker-1-general was increased to 1024 and layer FC-2 with another 1024 neurons was added. No dropout was applied. The following results were obtained: training loss = 1.0619e-04, validation loss = 8.5874e-05, training IoU = 0.88171, validation IoU = 0.88037.

## Segmentation module

The main features of the segmentation module are as follows:

1. **Segmenting an input patch.** We have trained the segmentation module to segment only a cell of interest within this patch, ignoring other cells, or fragments of other cells, present in the patch. After tracking, patches of size 96 x 96 centered on cell centroids (as obtained from the tracking module) are fed into the segmentation module. The segmented patch is then pasted into an empty, black image of size 382 x 382.

2. **Seed image.** This idea was illustrated in Fig 2B. A single seed is used to identify the cell of interest, which may not be in the centre of the patch.

3. **Novelty of the approach**. A similar approach was used in DeLTA [31] where four channels were used as inputs, including the segmented variant of the previous frame with only the cell of interest is left in the image. This segmented cell serves as a pointer to tell the U-Net which cell, out of the several present in the current frame, needs to be tracked. The main difference with our approach is that our pointers were artificially created. Also, in that paper, a pointer was used to choose a cell of interest from a segmented image. In our approach, we choose the cell first before segmenting.

4. **Classification task**. The task was cast as a classification at the pixel level, with the weighted cross-entropy loss function and IoU as the performance metric. The inputs to the segmentation module were 96 x 96 x 3 patches consisting of 3 channels: fluorescent, bright field and a seed channel. We added the bright field channel to the input as it allowed us to significantly improve the segmentation quality. The outputs are 96 x 96 binary patches where only a cell of interest determined by the seed channel is segmented, other cells or their fragments are ignored. To reinforce the learning process, we incorporated weight maps incorporated into the loss function. The weight maps for our task followed the method in [46] with pixels between close cells being assigned greater weights. Certain other pixels were also assigned higher weights, as explained below.

5. **Ensemble.** The segmentation module is an ensemble of two consecutive convolutional neural networks, Segmentor and Refiner, both having the U-Net architecture [46]. The Segmentor performs initial segmentation which is improved by the Refiner. Best results were achieved by adding outputs from both networks. The need for the ensemble approach became apparent after examining the types of errors occurring in the Segmentor. The IoU achieved on the validation set after applying trained Segmentor was modest (82%). A closer look at the segmented patches showed that errors occurred when T cells had particularly non-circular shapes (Fig 8A). Segmentor on its own would often break a cell into fragments. Refiner was designed to mitigate this problem and boosted the performance dramatically in terms of the averaged validation IoU (from 82% to 94%).

6. **Segmentor.** As described above, the role of the Segmentor is to perform initial segmentation of the cell of interest. Creating training data. An example of a training sample for Segmentor is shown in Fig 8B (top row): it consists of the input patch, the weight map, and the output (the label). The input patch includes 3 channels of size 96 x 96: the fluorescent channel, the bright field channel and the seed channel created artificially. The output (label) is the binary patch of the same size where only the cell of interest is segmented. Pixels values of input images were normalized per channel to a range between zero and one by subtracting the mean and dividing by the standard deviation, and segmented patches (labels) were normalised by dividing them by 255.

It is important to note that, to make the Segmentor more robust to tracking errors during execution, patches were randomly shifted relative to the actual centroids to mimic the inaccuracies of tracking. The random shifts for both X and Y coordinates of the centre of a patch were generated from the uniform distribution over the range (-35, 35). This introduced errors into the training data to replicate the imperfections and errors produced by the tracking module. Also, to reinforce the learning process, the weight maps were created for each training patch and then incorporated into the weighted cross-entropy loss function. As mentioned above, the formula for the weight maps was inspired by that in [46], where pixels between the close cells are assigned higher weights (see the first two terms in Formula (4)). In our approach, we also weighted pixels corresponding to the unwanted cells or their fragments. This yielded in the additional term $W_{fr}(\bar{x})$. The resulting formula for the weight map is

$$W_{Segmentor}(\bar{x}) = W_c(\bar{x}) + W_0 \cdot exp\left(-\frac{(d_1(\bar{x}) + d_2(\bar{x}))^2}{2\sigma^2}\right) + W_{fr}(\bar{x}) \qquad (4)$$

Here, $W_{fr}: R^{96 \times 96} \rightarrow R^{96 \times 96}$ is the weight map for unwanted fragments;
$\bar{x}$ are the coordinates of the $i$-th pixel in an image;
$W_c$, $d_1$ and $d_2$ are all functions over a two-dimensional image such that:
$W_c: R^{96 \times 96} \rightarrow R^{96 \times 96}$ is the class probability map.
$d_1: R^{96 \times 96} \rightarrow R^{96 \times 96}$ is the distance from the $i$-th pixel to the border of the nearest cell.
$d_2: R^{96 \times 96} \rightarrow R^{96 \times 96}$ is the distance from the $i$-th pixel to the border of the second nearest cell; $W_0$ and $\sigma$ are some constants. In [46], they were assigned values $W_0 = 10$ and $\sigma \approx 5$. We kept these values unchanged as they demonstrated the best performance during tuning process.

The value of $W_{fr}$ was set up equal to 8 for the pixels corresponding to the unwanted cells and $W_{fr} = 0$ for the rest. This value was chosen because it showed best performance during experimentation.

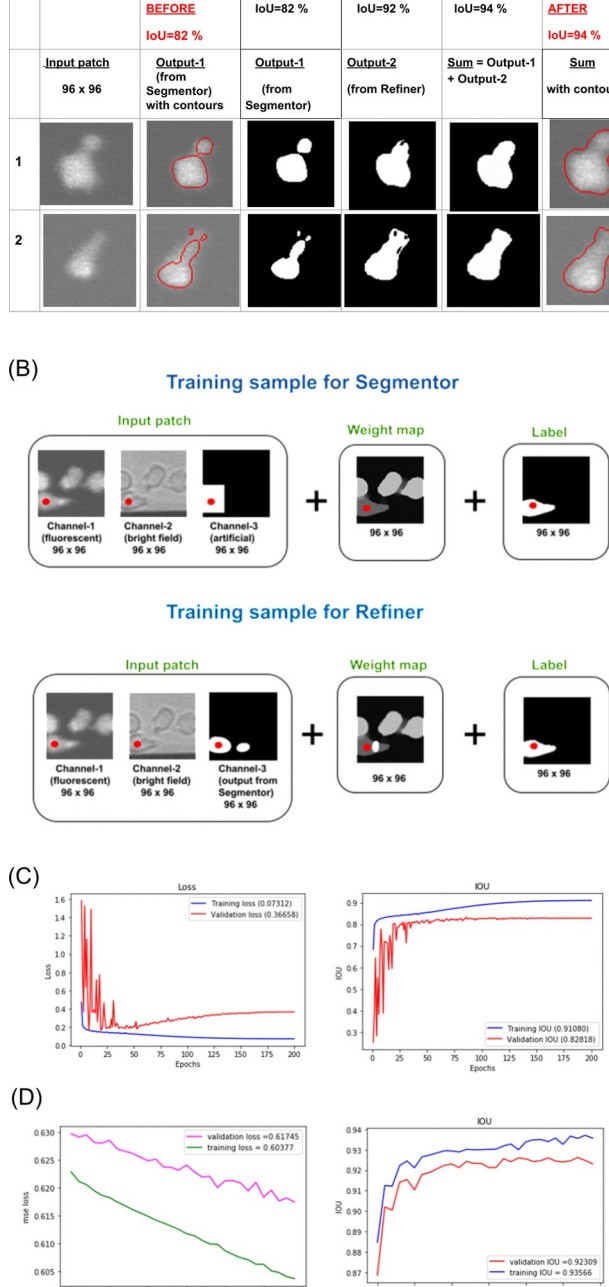

**Fig 8. Segmentation module training details.** (A) The results obtained from the Segmentation Ensemble. Each of the 2 rows displayed in this picture represent 2 different examples showing how implementing Segmentation Ensemble enabled a significant reduction in segmentation errors. The best results were achieved by adding outputs from Segmentor and Refiner together as can be observed from these images. In the second column (BEFORE) and the last one (AFTER), the segmentation results are shown as contours, plotted on top of the fluorescent images for visualization. Note that the IoU values given in the top row are not for these samples alone. They are average IoUs for the whole validation set. (B) Comparison of training samples for the Segmentor and the Refiner networks. This figure illustrates how the training data for Segmentor and Refiner was created. For both neural networks, the same set of input patches was utilized for training, but the seed channels and weight maps were created differently. The seed for the Segmentor was a 40 x 40 square located at the cell of interest (marked by the red dot) in the previous frame. The seed for the Refiner was the output from the Segmentor, i.e., the segmented cell of interest (with some potential errors). The weight maps for both neural networks contained higher weights for the unwanted cells, i.e., the cells or their fragments which were to be ignored, and for the pixels between close cells (this is visualized as regions with lighter grey

pixels). In addition, the weight map for the Refiner contains a bright region corresponding to the segmentation error produced by the Segmentor. (C) The learning curves for the Segmentor. *(Left)* The training and validation curves for the cross-entropy pixel wise loss function. (Right) The training and validation curves for the IoU (Intersection Over Union) metric. The model with the highest validation IoU (which corresponded to epoch 50) was saved. (D) The learning curves for the Refiner. (Left) Training and validation curves for the cross-entropy pixel wise loss function. Note that even though the validation curve is higher than the training curve, the absolute difference is small (0.014 at Epoch 30). (Right) Training and validation curves for the IoU (Intersection Over Union) metric. The model with the highest validation IoU was saved.

The validation data was created in the same fashion except for the weight maps, which were not needed at the validation stage. The validation samples consisted of 96 x 96 x 3 input images and 96 x 96 labels only.

Our modifications to the architecture. The U-Net model was downloaded from the GitHub repository: https://github.com/zhixuhao/unet/blob/master/model.py. Minor changes were made to some existing dropout layers, denoted as Drop4 and Drop5, where dropouts were increased from 0.3 to 0.5, and new ones, Drop2 = 0.5 and Drop3 = 0.3, were added to the convolutional layers Conv2 and Conv3. Also, because our input images were of size 96 x 96 x 3 (instead of 256 x 256 x 1 in the original model), we modified the size of the feature maps accordingly.

Training procedure. One of the distinctive properties of U-Net is that it requires a minimal amount of training data. The number of training samples was only 283 (although during training that number was increased 6-fold by augmentation: 3 rotations and 2 reflections) and the number of validation samples was 250. Training was performed for 200 epochs using the Adam optimiser with an initial learning rate of 0.00009 and a batch size of 32. The validation IoU achieved was 0.8282, i.e., 82%. Learning curves for both pixel-wise cross entropy loss and IoU are shown in Fig 8C. As can be observed around epoch number 50, the model started overfitting and the validation IoU curve plateaued. The model corresponding to the highest validation IoU (and, respectively, the lowest validation cross entropy loss) was saved.

**7. Refiner.** This is the second U-Net in the Segmentation Ensemble whose job is to refine results produced by the Segmentor. The final segmentation is obtained by adding the outputs from Segmentor and Refiner.

Creating training data. The difference between a training sample for the Refiner and a training sample for the Segmentor is explained in Fig 8B. Just as for Segmentor, the training sample for the Refiner comprises 3 parts: the input image, the label, and the weight map. The label is the same–the binary 96 x 96 image with the segmented cell of interest. The input consists of three 96 x 96 channels but the seed channel with a pointing square is replaced with the output from Segmentor, which now plays the role of the seed channel. Refiner compares the input from Segmentor and the label and makes corrections to segmentation if necessary. Refiner thus needs to see both the rough segmentation obtained from the Segmentor and the label during training to learn to detect and correct the difference. The Refiner is thus a seed-driven algorithm as well, like the trackers and Segmentor, although the seeds here are generated by Segmentor, rather than artificially.

To create training samples for Refiner, the following procedure was implemented. In the first step, all three components of the training samples for the Segmentor–the inputs (with seed channels), the weight maps and the labels were created. Then, the labels and the weight maps were set aside. Inputs were passed through the trained Segmentor thus yielding the segmented patches. This is the first approximation for segmentation. The seed channels in the inputs to Segmentor were substituted by these outputs from the Segmentor thus forming 96 x 96 x 3 input images for training the Refiner. Finally, the weight maps were created following the procedure detailed below.

The role of the weight map in this setting is to guide the learning process so that Refiner focuses on pixels that Segmentor failed to classify correctly. At the same time, since Refiner will face the same issues as Segmentor during training with class imbalance and close cells, the weights corresponding to those areas should remain unchanged. Thus, the only difference between Segmentor and Refiner weight maps is that, for Refiner, the pixels corresponding to the difference between the label and the segmentation obtained from Segmentor (and, therefore, representing the segmentation error produced by the Segmentor) should be assigned larger weights. This idea is illustrated in Fig 8A (bottom row): Segmentor failed to segment the uropod of the T cell (this can be deduced by looking at the label and comparing it with the output from the Segmentor). The corresponding region in the weight map is bright, indicating that these weights are much bigger than the rest.

Thus, the pixels which turn out to be difficult for Segmentor are assigned higher weights when creating training samples for the Refiner. To detect these pixels in each training sample, the output from the Segmentor was subtracted from its corresponding label and the absolute value of the difference was taken, i.e.

$$Difference = Abs(Output\ from\ Segmentor - Label) \tag{5}$$

As both terms on the right in (5 are binary images, the result is binary as well, where each pixel is either 0 or 255. The regions having intensities 0 in the Difference in (5 corresponded to the correct segmentation performed by the Segmentor while the regions with intensity 255 represented a segmentation error produced by the Segmentor which needed to be corrected by the Refiner.

After the regions of segmentation errors were detected, the weight maps $W_{errors}: R^{96\,x\,96} \rightarrow R^{96\,x\,96}$ were created where pixels corresponding to the regions of segmentation errors were assigned intensity 10, i.e. $W_{errors}(\bar{x}) = 10$ (this value was established as a result of tuning process) and finally, were added to the weight maps $W_{Segmentor}(\bar{x})$ built previously for the Segmentor according to Formula (4), i.e.

$$W_{Refiner}(\bar{x}) = W_{Segmentor}(\bar{x}) + W_{errors}(\bar{x}) \tag{6}$$

Architecture of the Refiner. The architecture of the initial U-Net downloaded from GitHub repository mentioned above, as well as in the case of the Segmentor, was not modified, only dropout layers Drop2 = 0.5 and Drop3 = 0.3 were added to the convolutional layers Conv2 and Conv3 to address overfitting during training.

Training procedure. As with Segmentor, pixel values of input images were normalized per channel to a range between zero and one by subtracting the mean and dividing by the standard deviation, while segmented patches (labels) were normalised by dividing them by 255. The number of training samples was 371 and 6-fold augmentation was applied; the number of validation samples was 250 (the same validation set used for Segmentor was utilised again). The Adam optimiser with the initial learning rate 0.00009 and batch size of 32 were utilized. The training was performed for 30 epochs only. This was enough to get a boost in IoU from 82% to 92%. Learning curves for both pixel-wise cross entropy loss and IoU are shown in Fig 8D.

## Division detector

The ability to detect cell divisions is important for DeepKymoTracker to run smoothly. As described above, each of the neural networks (Tracker-1, Tracker-2, . . ., Tracker-5) has been trained to track a fixed number of cells. It is thus essential to detect cell divisions since this is when we need to switch from one tracker to another.

Due to the uniform and repetitive nature of mitosis events in T cells, we were able to use classical computer vision techniques to detect cell division events. The Division Detector is applied to segmented patches returned by the Segmentation Ensemble and consists of 2 independent branches: a pre-division phase detector (which specialises in detecting the figure eight shape of a pre-mitotic cell by utilizing convexity defects in the cells' shapes to differentiate between figure eight shapes and other shapes) and a post-division phase detector (whose job is to detect two daughter cells both having a regular round shape with practically equivalent radii; this is done by comparing their radii and circularities).

## Author Contributions

**Conceptualization:** Sarah M. Russell, Damien G. Hicks.

**Data curation:** Kajal Zibaei, Mohammed Yassin.

**Formal analysis:** Khelina Fedorchuk, Mohammed Yassin.

**Funding acquisition:** Sarah M. Russell, Damien G. Hicks.

**Investigation:** Khelina Fedorchuk.

**Methodology:** Khelina Fedorchuk, Damien G. Hicks.

**Project administration:** Sarah M. Russell, Damien G. Hicks.

**Resources:** Kajal Zibaei, Mohammed Yassin.

**Software:** Khelina Fedorchuk.

**Supervision:** Sarah M. Russell, Damien G. Hicks.

**Validation:** Khelina Fedorchuk.

**Visualization:** Khelina Fedorchuk.

**Writing – original draft:** Khelina Fedorchuk.

**Writing – review & editing:** Khelina Fedorchuk, Sarah M. Russell, Damien G. Hicks.

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
