## [Decision Letter · Decision Letter 0]

7 Oct 2024

PONE-D-24-27905DeepKymoTracker: A tool for accurate construction of cell lineage trees for highly motile cellsPLOS ONE

Dear Dr. Hicks,

Thank you for submitting your manuscript to PLOS ONE. After careful consideration, we feel that it has merit but does not fully meet PLOS ONE’s publication criteria as it currently stands. Therefore, we invite you to submit a revised version of the manuscript that addresses the points raised during the review process.

The reviewers have suggested to improve the readability of the manuscript in terms of the literature survey, description of the experimental setup and discussion of the obtained results. The contributions of the conducted study have to be highlighted in the context of the relevant literature body as well.

We look forward to receiving your revised manuscript.

Kind regards,

Muhammad Bilal, Ph.D.

Academic Editor

PLOS ONE

Journal Requirements:

1. When submitting your revision, we need you to address these additional requirements. Please ensure that your manuscript meets PLOS ONE's style requirements, including those for file naming. The PLOS ONE style templates can be found at https://journals.plos.org/plosone/s/file?id=wjVg/PLOSOne_formatting_sample_main_body.pdf and https://journals.plos.org/plosone/s/file?id=ba62/PLOSOne_formatting_sample_title_authors_affiliations.pdf 2. Please update your submission to use the PLOS LaTeX template. The template and more information on our requirements for LaTeX submissions can be found at http://journals.plos.org/plosone/s/latex.

Reviewers' comments:

Reviewer's Responses to Questions

**Comments to the Author**

1. Is the manuscript technically sound, and do the data support the conclusions?

Reviewer #1: Yes

Reviewer #2: Yes

Reviewer #3: Partly

Reviewer #4: Yes

Reviewer #5: Partly

Reviewer #6: Yes

Reviewer #7: Yes

Reviewer #8: Yes

2. Has the statistical analysis been performed appropriately and rigorously? 

Reviewer #1: Yes

Reviewer #2: Yes

Reviewer #3: No

Reviewer #4: Yes

Reviewer #5: No

Reviewer #6: Yes

Reviewer #7: Yes

Reviewer #8: Yes

3. Have the authors made all data underlying the findings in their manuscript fully available?

Reviewer #1: Yes

Reviewer #2: Yes

Reviewer #3: No

Reviewer #4: Yes

Reviewer #5: No

Reviewer #6: Yes

Reviewer #7: Yes

Reviewer #8: Yes

4. Is the manuscript presented in an intelligible fashion and written in standard English?

Reviewer #1: Yes

Reviewer #2: Yes

Reviewer #3: No

Reviewer #4: Yes

Reviewer #5: No

Reviewer #6: Yes

Reviewer #7: Yes

Reviewer #8: Yes

5. Review Comments to the Author

Reviewer #1: 1- The research gap is not well explained in the abstract as well as the proposed algorithm steps.

2- Add keywords

3- The introduction should contain a general introduction to the field of work presented, followed by an explanation of the work presented and its tools before going directly into the details of the tools. Please reorganize the introduction according to what was mentioned.

4- The tools are presented before the results. The general structure of the research needs to be reorganized.

5- Compare the submitted work with other previous works.

6- Unify reference formatting

Reviewer #2: 1. Your abstract does not highlight the specifics of your research or findings. Rewrite the Abstract section to be more meaningful. I suggest to be Problem, Aim, Methods, Results, and Conclusion.

2. The paper must identify the gap that the research, which the authors conducted, fills.

3. Related work is too lengthy, but the more crucial issue is that the section does not have good organization.

4. More Keywords are required.

5. I feel that more explanation would be need on how the proposed method is performed.

6. How were these techniques chosen, and what advantages do they offer over alternative methods?

7. Performance improvements are substantial, with IoU increasing from 82% to 92%. Additional metrics like precision and recall would provide a more comprehensive evaluation.

8. The division detection method is practical and well-designed, but more examples or case studies showcasing its effectiveness would strengthen the results.

9. The availability of code, demo data, and pre-trained weights is excellent for reproducibility and accessibility.

10. The manuscript is generally clear and well-organized. Improving figure legends and ensuring consistency in terminology would enhance readability.

11. Though the results are interesting, what scientific problems, can these results reveal?

12. Rewrite the Conclusion section to be:

- You must more clearly highlight the theoretical and practical implications of your research

-Discuss research contributions.

-Indicate practical advantages (in at least one separate paragraph),

-discuss research limitations (at least one separate paragraph), and

-supply 2-3 solid and insightful future research suggestions.

13. The references are relevant and comprehensive but could benefit from including more recent studies to provide a current context. The related work and the references are very old, and it seems that authors need to refer to latest work to justify the current approach.

14. Recommendations:

Include additional performance metrics like precision, recall, and F1 score.

Compare the proposed method with state-of-the-art techniques.

Add examples or case studies for the Division Detector.

Improve figure legends and ensure consistency in terminology.

Reviewer #3: The manuscript entitled “DeepKymoTracker: A tool for accurate construction of cell lineage trees for highly motile cells” has been investigated in detail. The paper presents DeepKymoTracker, a promising tool for the combined tracking and segmentation of motile cells like T lymphocytes, utilizing a novel seed-based mechanism and a 3D CNN. However, the explanation of core components like the seed mechanism and CNN architecture lacks depth, and the biological significance of the tool is underexplored. Additionally, benchmarking claims are not substantiated with enough quantitative evidence, and the generalizability of the method beyond T lymphocytes is not discussed. Revisions should focus on clarifying the technical innovations, providing more comprehensive results, and discussing the broader biological impact of the tool.

1) The introduction of the problem—tracking and segmenting highly motile cells like T lymphocytes—lacks clarity regarding the broader biological and medical relevance of cell lineage tracking. The paper should provide a stronger justification for why accurate cell tracking is critical in the context of T lymphocyte research, emphasizing its implications for understanding immune responses or other relevant biological processes.

2) The core innovation of DeepKymoTracker, the "seed" mechanism, is not adequately explained. How exactly does the seed transmit information between images and prevent tracking errors? The authors need to offer a more detailed, technical explanation of the seed’s role in both tracking and segmentation. Illustrations or diagrams of the seed-based process would help clarify this concept.

3) The description of the 3D convolutional neural network (CNN) used in the model is too vague. Critical details, such as the architecture, number of layers, types of activations, and training parameters, are missing. The authors should provide a more in-depth technical discussion about the design of the CNN and how it was optimized for the task of cell tracking.

4) While DeepKymoTracker is designed for T lymphocytes, the authors do not discuss whether the method generalizes to other cell types with different motility patterns or morphological characteristics. The applicability of this tool to a wider range of biological systems should be explored or at least discussed, particularly with regard to cells that may exhibit different behaviors.

5) The training process is said to involve synthetic and experimental T lymphocyte images, but there is no discussion about how well the synthetic data reflects real-world variability. Were steps taken to avoid overfitting to synthetic data? The authors should also clarify how the balance between synthetic and real data was handled and whether cross-validation was used to ensure robustness.

6) The paper highlights that DeepKymoTracker reduces the risk of identity errors, but there is little discussion on the types of errors that remain or any failure cases encountered during testing. The authors should present more comprehensive error analysis, including examples of scenarios where the tool struggles or where it makes incorrect predictions.

7) “Discussion” section should be edited in a more highlighting, argumentative way. The author should analysis the reason why the tested results is achieved.

8) While the tool is written in Python, no discussion is provided on its computational efficiency, scalability, or memory usage. How does DeepKymoTracker perform with large datasets or long time-lapse sequences? The authors should provide benchmarks for runtime performance, especially since time-lapse microscopy often generates large volumes of data.

9) The authors should clearly emphasize the contribution of the study. Please note that the up-to-date of references will contribute to the up-to-date of your manuscript. The study named- “Artificial intelligence-based robust hybrid algorithm design and implementation for real-time detection of plant diseases in agricultural environments”- can be used to explain the methodology and highlight the performance in the study or to indicate the contribution in the “Introduction” section.

10) Although the paper is focused on computational methods, there is little discussion about the biological insights gained from improved tracking. How does DeepKymoTracker contribute to a better understanding of T lymphocyte behavior? The authors should highlight specific biological findings or interpretations that were made possible due to the improved accuracy of the tool.

Reviewer #4: **General Comments:**

1. **Technical Soundness:** The manuscript is technically sound, and the methodology is well-developed. The integration of a 3D convolutional neural network for tracking highly motile cells and the seed-based approach for segmentation are well-explained and validated. The benchmarks against other tools are rigorous and demonstrate clear improvements in segmentation and tracking accuracy.

2. **Statistical Analysis:** The statistical analysis has been performed appropriately and rigorously. Key performance metrics such as precision, recall, F1-score, and IoU thresholds have been used, and the benchmarking against publicly available tools adds credibility to the findings. The inclusion of detailed tables and graphs enhances the robustness of the results.

3. **Data Availability:** The authors have made the code, data, and pre-trained models available on GitHub, which demonstrates a commitment to open science and reproducibility. This transparency is commendable.

4. **Clarity and Language:** The manuscript is written in clear, standard English and is easy to follow. The descriptions of complex techniques, like seed-based tracking and the CNN architecture, are well-articulated. The authors have successfully communicated their contributions to both the cell-tracking and machine-learning communities.

---

**Specific Comments:**

1. **Introduction:** The introduction effectively sets the context for the problem of tracking highly motile cells like T lymphocytes. However, expanding the discussion on why previous approaches struggle with these cells could strengthen the motivation for the new method.

2. **Methodology:** The pipeline for the DeepKymoTracker is well-explained, but a more detailed explanation of how synthetic data was generated could be helpful. Since synthetic data plays a critical role in training the network, a brief discussion on potential limitations or biases from synthetic data would improve the transparency of the approach.

3. **Figures and Tables:** The figures and tables are clear and provide valuable insight into the performance of the proposed method. However, in Figure 4, providing more discussion on why performance degrades for higher IoU thresholds would be helpful, especially in explaining the challenges with more complex cell shapes.

4. **Discussion:** The discussion section is thorough, but it would benefit from a more in-depth exploration of the potential future applications of DeepKymoTracker beyond the scope of T cells. This could include tracking other highly motile cells or expanding the method to work with different microscopy techniques.

5. **Execution Time:** While the performance of DeepKymoTracker is clearly demonstrated, it would be useful to include a brief mention of computational resource requirements, particularly for users who may not have access to high-performance computing systems.

---

**Conclusion:** Overall, this manuscript is a valuable contribution to the field of cell tracking. It introduces a novel method that addresses key challenges in tracking highly motile cells and provides a robust solution with clear improvements over existing tools. The availability of the code and data also promotes further research and application.

**Recommendation: Minor Revision**

While the manuscript is technically sound and well-written, there are a few areas where clarification and additional details could improve the overall quality. These include expanding on the generation of synthetic data, providing more discussion on potential limitations, and offering further insights into specific performance metrics. These issues are relatively minor and should not require significant changes to the core methodology or results. Therefore, a **minor revision** would be appropriate.

Reviewer #5: The DeepKymoTracker paper has several substantial defects despite its innovative approach to monitoring and segmenting motile cells through microscopy. The logic from the problem statement to the suggested solution and confirmation is complex due to the disconnect between technique descriptions and theoretical underpinning.

Using 3D CNNs, seed-based tracking and segmentation improves originality. However, this approach must continue to be regarded as adventurous. CNNs with tracking-by-detection paradigms have been employed in the literature for an extended period. The manuscript fails to differentiate the tool from existing methods or illustrate how the integration of seeds into the monitoring mechanism represents a significant advancement.

The authors recognize the challenge of identifying exceedingly motile cells, but they do not specify the research gap that their method addresses. The unique contribution of DeepKymoTracker and the broader context of cell-tracking research are not thoroughly highlighted.

Using seeds to mitigate identity shifts between time-lapse frames is intriguing regarding creativity and additionality; however, it has yet to achieve revolutionary status. The manuscript improves upon existing methodologies rather than introducing a new theory or paradigm. The paper requires a more robust argument to support the assertion that DeepKymoTracker advances methodology. The literature review enumerates existing methodologies; however, further critical analysis is required. Descriptive discussions of related works provide minimal analysis or comparison to elucidate how the new work differs from or expands upon previous endeavors. This section will be enhanced by conducting a critical analysis of previous methods' deficiencies and how this new tool addresses them. Although the approach section is extensive, it must clearly demonstrate its value. The integration of monitoring and segmentation is meticulously described; however, the advantages of alternative solutions are not. The technical aspects are overemphasized by the authors, who neglect to consider the impact of their methodological decisions on the efficacy and accuracy of the reported results. The results are readily apparent; however, statistical rigor is required. The actual positive rate and intersection-over-union (IoU) are provided; however, the statistical analysis does not extend beyond reporting results. Measurements are compared to other instruments without statistical testing to determine significance. Examine the model's generalizability to various datasets and reproducibility of the results. The discussion must situate the study's findings within a broader context of research. A comprehensive comparative analysis must demonstrate that their approach surpasses current technologies.

Conversely, they assert that DeepKymoTracker has superior performance without explaining the reasons for its success in specific domains or confronting potential limitations. In-depth summaries of the study's findings should be included in the conclusion. The authors must provide a detailed explanation of the practical applications of the instrument for scientists. The results do not include a forward-thinking discussion about the instrument's potential enhancements or applications in novel situations. The article introduces a functional cell monitoring tool; however, it is structurally flawed, lacks innovation, and lacks methodological or statistical robustness to significantly contribute to the field. It requires a greater emphasis on DeepKymoTracker's practical implications and competitive advantages, as well as a more comprehensive theoretical framework. The paper's results could be improved with cross-validation or diverse dataset testing. Furthermore, authors need to analyze prior research and consider how their work outperforms them throughout the discussion.

Reviewer #6: The manuscript presents a proposed method for combining tracking and segmentation for highly motile cells. The manuscript is technically sound and presented in good fashion. It shows a promising results in comparison with other benchmarks.

Reviewer #7: Overall Impression:

The manuscript presents DeepKymoTracker, a novel tool integrating 3D Convolutional Neural Networks (CNNs) for the accurate tracking and segmentation of highly motile cells, specifically T lymphocytes, in time-lapse microscopy. The innovative use of seed-driven mechanisms and CNNs to detect and associate cells across consecutive images represents a significant advancement in cell tracking, outperforming several existing tools. The software shows promise in addressing challenges related to cell misidentification and lineage assembly, which are crucial for studies on heritability and cell decision-making.

Strengths:

Novel Contribution: The paper introduces a new integrated pipeline that merges tracking and segmentation with high accuracy, particularly using 3D CNNs. The novel seed-driven mechanism ensures that cell identity is preserved across frames, minimizing misidentification.

Benchmarking & Comparison: The tool was rigorously benchmarked against five publicly available tools and demonstrated superior performance across key measures such as segmentation, detection, and tracking. The comprehensive comparison enhances the credibility of the results.

Technical Rigor: The paper provides detailed explanations of the model architectures (both for tracking and segmentation), training procedures, and performance metrics. The clarity in the presentation of algorithms and their corresponding validation on both synthetic and real-world datasets adds substantial weight to the research.

Reproducibility: The open-source nature of the tool, with the code and pre-trained models made freely available, supports the reproducibility and accessibility of the research.

Suggestions for Improvement:

Generalizability Across Cell Types: While the tool is validated on T lymphocytes, it would be beneficial to explore its application on other cell types. The manuscript could expand its scope by including additional datasets or experimental results that demonstrate DeepKymoTracker's adaptability to other highly motile cells in various environments.

Real-world Dataset Limitations: A portion of the training data for some trackers (e.g., Tracker-3, Tracker-4, and Tracker-5) relies on synthetic data. While this approach was effective, expanding the real-world training and validation datasets would further substantiate the tool's effectiveness in practical applications.

Ease of Use and Deployment: Although the technical depth of the paper is commendable, more attention could be given to the usability of the tool. A section detailing the computational requirements, installation procedures, and expected performance on different hardware configurations (beyond execution times) could enhance the manuscript’s appeal to a broader audience in both computational and biological labs.

Visual Demonstrations: The visual output of the tool is compelling, and the inclusion of more high-resolution visualizations in the manuscript could greatly benefit the readers. More figures demonstrating the comparison between the performance of DeepKymoTracker and other tools would enhance clarity and impact.

Discussion on Scalability: A more detailed discussion on the scalability of the tool for larger datasets with higher cell densities could be added. This is especially important if the tool is to be applied in large-scale experiments or on high-throughput platforms.

Minor Comments:

Clarity on Training Data: It would be helpful to provide more information regarding the balance between synthetic and real-world data used in training each tracker, along with a clearer justification for the choice of synthetic data percentages.

Typographical Corrections: There are a few minor grammatical and typographical errors throughout the manuscript that should be addressed in the final version.

Figures: Ensure all figures have consistent formatting, and consider adding more visual examples of tracking errors and how DeepKymoTracker corrects them.

Reviewer #8: Review of the article "DeepKymoTracker: A tool for accurate construction of cell lineage trees for highly motile cells."

The authors propose developing DeepKymoTracker, a methodology for combined tracking and segmenting cells under high movement, constructing a lineage tree to identify a hierarchy in the cell division process.

They used the concept of seed channel to correct the tracking and segmentation steps. The results show that the approach is very convenient. The authors also tested the methodology in several cell movies and compared it with other similar solutions, reaching good numbers in the measurements used by the authors. So, I believe the article is worth publishing in the journal.

Nonetheless, I suggest including some information about the method used to construct the seed regions to enhance the article. It is unclear how the seeds are produced at every start of clip processing and if they are produced automatically. Also, include some implementation details.

It is well written, but the English style could be improved. Also, there are some typos in the text.

6. PLOS authors have the option to publish the peer review history of their article (what does this mean?). If published, this will include your full peer review and any attached files.

Reviewer #1: No

Reviewer #2: **Yes: **Waqar Ahmad

Reviewer #3: No

Reviewer #4: No

Reviewer #5: No

Reviewer #6: No

Reviewer #7: **Yes: **ZULFIKAR ALI ANSARI

Reviewer #8: **Yes: **Leopoldo Altamirano Robles

---

## [Author Response · Author response to Decision Letter 0]

21 Nov 2024

EDITORS` SUGGESTIONS:

1. The reviewers have suggested to improve the readability of the manuscript in terms of the literature survey, description of the experimental setup and discussion of the obtained results. The contributions of the conducted study have to be highlighted in the context of the relevant literature body as well.

• RESPONSE: 

 We have updated the literature review (see the Introduction section and References). We have also enhanced the description of the experimental setup by adding more details about how the synthetic data was created, how the seeds transmit information and how they are automatically created during execution (see Materials and Methods). We have reorganized the Introduction section to emphasize our contributions more explicitly compared to the previous works. Finally, we have included discussion of the obtained results in terms of their applicability to other cell types (see the Discussion section).

REVIEWER 1 COMMENTS:

2. The research gap is not well explained in the abstract as well as the proposed algorithm steps.

• RESPONSE: We have explicitly stated the research gap at the end of the Introduction.

3. Add keywords

• RESPONSE: We have added keywords.

4. The introduction should contain a general introduction to the field of work presented, followed by an explanation of the work presented and its tools before going directly into the details of the tools. Please reorganize the introduction according to what was mentioned

• RESPONSE: We have reorganized the Introduction accordingly.

5. The tools are presented before the results. The general structure of the research needs to be reorganized.

• RESPONSE: The tools are presented briefly in order to provide context for the results. Most detail about the tools are presented in the Materials and Methods. 

6. Compare the submitted work with other previous works.

• RESPONSE: We have now extensively compared our approach with other previous methods in the Introduction section. In addition, as we describe each step of our method in the Materials and Methods section, we reference methods that we build on or that are related. 

7. Unify reference formatting

• RESPONSE: We have made sure that the reference formatting is consistent throughout the paper. 

REVIEWER 2 COMMENTS:

8. . Your abstract does not highlight the specifics of your research or findings. Rewrite the Abstract section to be more meaningful. I suggest to be Problem, Aim, Methods, Results, and Conclusion.

• RESPONSE: We have confirmed that our abstract follows this template:

1. Problem - the first 5 sentences (until “To address this problem…”) describes the problem.

2. Aim - the sentence “To address this problem…”. 

3. Methods – from” Central to DeepKymoTracker…” to” It was benchmarked …”

4. Results -from “It was benchmarked…” to “freely available”.

5. Conclusion – “We suggest this tool is particularly suited….”.

9. The paper must identify the gap that the research which the authors conducted fills.

• RESPONSE: We have explicitly stated the research gap at the end of the Introduction.

10. Related work is too lengthy, but the more crucial issue is that the section does not have good organization.

• RESPONSE: We have reorganized the introduction to be clearer. 

11. More Keywords are required.

• RESPONSE: We have added more keywords.

12. . I feel that more explanation would be need on how the proposed method is performed.

• RESPONSE: We have added more material on how exactly seeds transmit information between the modules (see Principles of Tracking Module, Paragraph 5 and Principles of Segmentation Module, Paragraph 4). 

13. How were these techniques chosen, and what advantages do they offer over alternative methods?

• RESPONSE: We have discussed the advantages and limitations of our method in the Discussion.

14. . Performance improvements are substantial, with IoU increasing from 82% to 92%. Additional metrics like precision and recall would provide a more comprehensive evaluation.

• RESPONSE: In addition to IoU, we provided DET, TRA and SEG measures from Cell Tracking Challenge which are currently widely accepted by the cell tracking community. Unfortunately, it was not realistic to rerun all 5 tools to extract precision and recall in the given time frame for revision. 

15. The division detection method is practical and well-designed, but more examples or case studies showcasing its effectiveness would strengthen the results.

• RESPONSE: The division detection step was not the main topic of our research and we only touched on it briefly in our paper. In the discussion we have included a comment on how division detection might be incorporated into detection of occlusions and cell death in a future improvement.

16. The availability of code, demo data, and pre-trained weights is excellent for reproducibility and accessibility.

• RESPONSE: Thank you very much for the positive feedback.

17. The manuscript is generally clear and well-organized. Improving figure legends and ensuring consistency in terminology would enhance readability.

• RESPONSE: Thank you very much.

18. Though the results are interesting, what scientific problems can these results reveal?

• RESPONSE: We have discussed some of the scientific problems in our revised Introduction.

19. . Rewrite the Conclusion section to be:

- You must more clearly highlight the theoretical and practical implications of your research

-Discuss research contributions.

-Indicate practical advantages (in at least one separate paragraph),

-discuss research limitations (at least one separate paragraph), and

-supply 2-3 solid and insightful future research suggestions.

• RESPONSE: We have added this information (see Discussion, the last two paragraphs). As regards the research contributions, they are now highlighted more explicitly in the Introduction section.

20. The references are relevant and comprehensive but could benefit from including more recent studies to provide a current context. The related work and the references are very old, and it seems that authors need to refer to latest work to justify the current approach

• RESPONSE: We have updated the references to include more recent citations (see Introduction and References).

21. Recommendations:

Include additional performance metrics like precision, recall, and F1 score.

Compare the proposed method with state-of-the-art techniques.

Add examples or case studies for the Division Detector.

Improve figure legends and ensure consistency in terminology.

• RESPONSE: We have responded to these points above. 

REVIEWER 3 COMMENTS:

22. The manuscript entitled “DeepKymoTracker: A tool for accurate construction of cell lineage trees for highly motile cells” has been investigated in detail. The paper presents DeepKymoTracker, a promising tool for the combined tracking and segmentation of motile cells like T lymphocytes, utilizing a novel seed-based mechanism and a 3D CNN. However, the explanation of core components like the seed mechanism and CNN architecture lacks depth, and the biological significance of the tool is underexplored. Additionally, benchmarking claims are not substantiated with enough quantitative evidence, and the generalizability of the method beyond T lymphocytes is not discussed. Revisions should focus on clarifying the technical innovations, providing more comprehensive results, and discussing the broader biological impact of the tool.

• RESPONSE: We have now added more information on the seed mechanism and the generalizability of the method (see our responses in the comments below). 

23. The introduction of the problem—tracking and segmenting highly motile cells like T lymphocytes—lacks clarity regarding the broader biological and medical relevance of cell lineage tracking. The paper should provide a stronger justification for why accurate cell tracking is critical in the context of T lymphocyte research, emphasizing its implications for understanding immune responses or other relevant biological processes.

• RESPONSE: We have added this information in the Introduction section.

24. The core innovation of DeepKymoTracker, the "seed" mechanism, is not adequately explained. How exactly does the seed transmit information between images and prevent tracking errors? The authors need to offer a more detailed, technical explanation of the seed’s role in both tracking and segmentation. Illustrations or diagrams of the seed-based process would help clarify this concept.

• RESPONSE: We have added this information (see Principles of Tracking Module, Paragraph 5 and Principles of Segmentation Module, Paragraph 4). 

25. The description of the 3D convolutional neural network (CNN) used in the model is too vague. Critical details, such as the architecture, number of layers, types of activations, and training parameters, are missing. The authors should provide a more in-depth technical discussion about the design of the CNN and how it was optimized for the task of cell tracking.

• RESPONSE: The architecture was described in the Tracking Module section (Materials and Methods) and shown in Figure 5 A as well as in Table 7 and Table 8. The training parameters were listed in Table 6. The types of activations were described in the same section.

26. While DeepKymoTracker is designed for T lymphocytes, the authors do not discuss whether the method generalizes to other cell types with different motility patterns or morphological characteristics. The applicability of this tool to a wider range of biological systems should be explored or at least discussed, particularly with regard to cells that may exhibit different behaviors.

• RESPONSE:. We have added this information to the discussion

27. The training process is said to involve synthetic and experimental T lymphocyte images, but there is no discussion about how well the synthetic data reflects real-world variability. Were steps taken to avoid overfitting to synthetic data? The authors should also clarify how the balance between synthetic and real data was handled and whether cross-validation was used to ensure robustness.

• RESPONSE:. We have provided information on the simulation of and training with synthetic data see 3. Synthetic data (Tracking Module, in Materials and Methods)).

28. The paper highlights that DeepKymoTracker reduces the risk of identity errors, but there is little discussion on the types of errors that remain or any failure cases encountered during testing. The authors should present more comprehensive error analysis, including examples of scenarios where the tool struggles or where it makes incorrect predictions.

• RESPONSE: We have included this information in the Discussion section (about divisions, occlusions, and dying cells).

29. “Discussion” section should be edited in a more highlighting, argumentative way. The author should analysis the reason why the tested results is achieved.

• RESPONSE: We have provided some information on the performance of our tool on other cell types.

30. While the tool is written in Python, no discussion is provided on its computational efficiency, scalability, or memory usage. How does DeepKymoTracker perform with large datasets or long time-lapse sequences? The authors should provide benchmarks for runtime performance, especially since time-lapse microscopy often generates large volumes of data.

• RESPONSE: We gave comparative execution times of DeepKymoTracker on different machines in Table 5. 

31. The authors should clearly emphasize the contribution of the study. Please note that the up-to-date of references will contribute to the up-to-date of your manuscript. The study named- “Artificial intelligence-based robust hybrid algorithm design and implementation for real-time detection of plant diseases in agricultural environments”- can be used to explain the methodology and highlight the performance in the study or to indicate the contribution in the “Introduction” section.

• RESPONSE: The mentioned study was helpful for organizing our Introduction.

32. Although the paper is focused on computational methods, there is little discussion about the biological insights gained from improved tracking. How does DeepKymoTracker contribute to a better understanding of T lymphocyte behavior? The authors should highlight specific biological findings or interpretations that were made possible due to the improved accuracy of the tool.

RESPONSE: We have provided more context for T cell biology in the Introduction. 

REVIEWER 4 COMMENTS:

33. **Introduction:** The introduction effectively sets the context for the problem of tracking highly motile cells like T lymphocytes. However, expanding the discussion on why previous approaches struggle with these cells could strengthen the motivation for the new method

• RESPONSE: We have rewritten the Introduction section putting more emphasis on this problem (we highlighted more explicitly the fact that, unlike in the previous works, we managed to perform detection and association simultaneously which allowed us to reduce the number of tracking errors occurring during the association step).

34. **Methodology:** The pipeline for the DeepKymoTracker is well-explained, but a more detailed explanation of how synthetic data was generated could be helpful. Since synthetic data plays a critical role in training the network, a brief discussion on potential limitations or biases from synthetic data would improve the transparency of the approach

• RESPONSE: We have described synthetic data implementation in detail now.

35. **Figures and Tables:** The figures and tables are clear and provide valuable insight into the performance of the proposed method. However, in Figure 4, providing more discussion on why performance degrades for higher IoU thresholds would be helpful, especially in explaining the challenges with more complex cell shapes.

• RESPONSE: We have explained the reason in the caption to Figure 4 and in the text below.

36. **Discussion:** The discussion section is thorough, but it would benefit from a more in-depth exploration of the potential future applications of DeepKymoTracker beyond the scope of T cells. This could include tracking other highly motile cells or expanding the method to work with different microscopy techniques.

• RESPONSE: We have added this information to the Discussion.

37. **Execution Time:** While the performance of DeepKymoTracker is clearly demonstrated, it would be useful to include a brief mention of computational resource requirements, particularly for users who may not have access to high-performance computing systems.

• RESPONSE: We have provided a table of execution times for different machines in Table 5.

38. **Recommendation: Minor Revision**

While the manuscript is technically sound and well-written, there are a few areas where clarification and additional details could improve the overall quality. These include expanding on the generation of synthetic data, providing more discussion on potential limitations, and offering further insights into specific performance metrics. These issues are relatively minor and should not require significant changes to the core methodology or results. Therefore, a **minor revision** would be appropriate

• RESPONSE: We have addressed these points in the specific comments above.

REVIEWER 5 COMMENTS:

39. The DeepKymoTracker paper has several substantial defects despite its innovative approach to monitoring and segmenting motile cells through microscopy. The logic from the problem statement to the suggested solution and confirmation is complex due to the disconnect between technique descriptions and theoretical underpinning.

• RESPONSE: We address the specific points below.

40. Using 3D CNNs, seed-based tracking and segmentation improves originality. However, this approach must continue to be regarded as adventurous. CNNs with tracking-by-detection paradigms have been employed in the literature for an extended period. The manuscript fails to differentiate the tool from existing methods or illustrate how the integration of seeds into the monitoring mechanism represents a significant advancement.

• RESPONSE: In the Introduction we state that our motivation to combine detection and association is to reduce the association errors.. Re

---

## [Decision Letter · Decision Letter 1]

4 Dec 2024

DeepKymoTracker: A tool for accurate construction of cell lineage trees for highly motile cells

PONE-D-24-27905R1

Dear Dr. Hicks,

We’re pleased to inform you that your manuscript has been judged scientifically suitable for publication and will be formally accepted for publication once it meets all outstanding technical requirements.

Kind regards,

Muhammad Bilal, Ph.D.

Academic Editor

PLOS ONE

Additional Editor Comments (optional):

Reviewers' comments:

Reviewer's Responses to Questions

**Comments to the Author**

1. If the authors have adequately addressed your comments raised in a previous round of review and you feel that this manuscript is now acceptable for publication, you may indicate that here to bypass the “Comments to the Author” section, enter your conflict of interest statement in the “Confidential to Editor” section, and submit your "Accept" recommendation.

Reviewer #1: All comments have been addressed

Reviewer #2: All comments have been addressed

Reviewer #4: All comments have been addressed

Reviewer #7: All comments have been addressed

2. Is the manuscript technically sound, and do the data support the conclusions?

Reviewer #1: (No Response)

Reviewer #2: Yes

Reviewer #4: Yes

Reviewer #7: Yes

3. Has the statistical analysis been performed appropriately and rigorously? 

Reviewer #1: (No Response)

Reviewer #2: Yes

Reviewer #4: Yes

Reviewer #7: Yes

4. Have the authors made all data underlying the findings in their manuscript fully available?

Reviewer #1: (No Response)

Reviewer #2: Yes

Reviewer #4: Yes

Reviewer #7: Yes

5. Is the manuscript presented in an intelligible fashion and written in standard English?

Reviewer #1: (No Response)

Reviewer #2: Yes

Reviewer #4: Yes

Reviewer #7: Yes

6. Review Comments to the Author

Reviewer #1: (No Response)

Reviewer #2: I recommend accepting the manuscript. While the authors have made substantial improvements, I would suggest that the authors consider including precision and recall metrics in future work or whenever possible, as this would provide a more comprehensive evaluation of their method.

Reviewer #4: **Review Comments to the Author:**

1. **Innovative Approach**:

The DeepKymoTracker presents a compelling and innovative methodology for tracking and segmenting highly motile cells, particularly T lymphocytes. Its integration of a 3D CNN with seed-based tracking and segmentation is a valuable contribution to cell lineage tracking.

2. **Clarity in Methodology**:

While the manuscript provides a detailed explanation of the DeepKymoTracker's tracking and segmentation modules, certain aspects, such as the seed mechanism and its role in mitigating identity errors, could be elaborated further for better clarity.

3. **Generalizability**:

The study primarily focuses on T lymphocytes. It would benefit from discussing the tool's applicability to other cell types or biological contexts, providing a broader scope for its utility.

4. **Performance Metrics**:

The inclusion of additional metrics such as precision, recall, and F1 score enhances the robustness of the evaluation. However, statistical analysis comparing the tool's performance against others should be expanded to underline its effectiveness convincingly.

5. **Practical Applications**:

A more detailed discussion on the biological insights gained through improved tracking accuracy would strengthen the practical implications of the tool.

6. **Computational Efficiency**:

While execution times are provided, additional benchmarks for resource usage (memory and CPU/GPU requirements) would be valuable, particularly for users with limited computational infrastructure.

7. **Figures and Illustrations**:

The manuscript would benefit from high-resolution visualizations comparing DeepKymoTracker's output with other methods. Additionally, providing examples of typical tracking errors and how the tool resolves them would enhance its impact.

8. **Future Directions**:

Suggestions for future research, such as expanding the method to different microscopy techniques or addressing challenges in higher cell density environments, would offer a forward-looking perspective.

9. **Typographical Corrections**:

Minor grammatical and typographical errors should be addressed to improve readability.

Reviewer #7: Thanks for addressing all the comments. No further modifications are required. You can go for further process. Good Luck.

7. PLOS authors have the option to publish the peer review history of their article (what does this mean?). If published, this will include your full peer review and any attached files.

Reviewer #1: No

Reviewer #2: **Yes: **Waqar Ahmad

Reviewer #4: No

Reviewer #7: **Yes: **ZULFIKAR ALI ANSARI

---

## [Editor Report · Acceptance letter]

17 Jan 2025

PONE-D-24-27905R1 

PLOS ONE

Dear Dr. Hicks, 

I'm pleased to inform you that your manuscript has been deemed suitable for publication in PLOS ONE. Congratulations! Your manuscript is now being handed over to our production team.

Kind regards, 

on behalf of

Dr. Muhammad Bilal 

Academic Editor

PLOS ONE